# Aβ receptors specifically recognize molecular features displayed by fibril ends and neurotoxic oligomers

Ladan Amin[1] & David A. Harris [1✉]

Several cell-surface receptors for neurotoxic forms of amyloid-β (Aβ) have been described, but their molecular interactions with Aβ assemblies and their relative contributions to mediating Alzheimer's disease pathology have remained uncertain. Here, we used super-resolution microscopy to directly visualize Aβ-receptor interactions at the nanometer scale. We report that one documented Aβ receptor, PrP$^C$, specifically inhibits the polymerization of Aβ fibrils by binding to the rapidly growing end of each fibril, thereby blocking polarized elongation at that end. PrP$^C$ binds neurotoxic oligomers and protofibrils in a similar fashion, suggesting that it may recognize a common, end-specific, structural motif on all of these assemblies. Finally, two other Aβ receptors, FcγRIIb and LilrB2, affect Aβ fibril growth in a manner similar to PrP$^C$. Our results suggest that receptors may trap Aβ oligomers and protofibrils on the neuronal surface by binding to a common molecular determinant on these assemblies, thereby initiating a neurotoxic signal.

[1] Department of Biochemistry, Boston University School of Medicine, Boston, MA, USA. ✉email: daharris@bu.edu

Alzheimer's disease (AD) is a progressive neurodegenerative disease that is characterized by accumulation within the brain of extracellular plaques composed of the amyloid-β (Aβ) peptide, and intracellular neurofibrillary tangles containing abnormally phosphorylated forms of the tau protein[1,2]. These two kinds of pathological deposits lead ultimately to synaptic loss and dysfunction, and to degeneration and loss of neurons. Aβ peptides of 40–42 amino acids in length are derived from the amyloid precursor protein (APP) via sequential cleavage by the enzymes β- and γ-secretase[2,3]. In AD brain, Aβ is either over-produced and/or degraded inefficiently, resulting in the formation of several types of Aβ aggregates[4]. There is strong evidence that small Aβ oligomers (Aβo), rather than monomers or fibrils, represent the key neurotoxic species in AD[5–7]. It is presumed that the disease process starts by the binding of Aβo to receptor proteins or lipids on the surface of neurons. However, the molecular identity of the relevant binding sites, and the signal transduction pathways they trigger leading to synaptotoxicity, are uncertain. Identification of Aβ receptors and elucidation of their mechanism of interaction with Aβo have important therapeutic implications, since these receptors represent potential pharmacological targets for the treatment of AD.

A number of cell-surface proteins have been reported to act as Aβ receptors[8,9]. Among them, the cellular prion protein (PrP$^C$)[10], Fcγ receptor IIb (FcγRIIb)[11], leukocyte immunoglobulin-like receptor; subfamily B2 (LilrB2)[12], ephrin type-B receptor 2 (EphB2)[13], and Nogo receptor family (Ngr1-3)[14], have attracted particular attention because of their high affinity for Aβo, and their ability to transduce a neurotoxic signal. However, there has been considerable controversy about the relative importance of each of these receptors in mediating Aβ neurotoxicity, with discrepant results emanating from many of the published studies[15–18]. At this point, it seems reasonable to assume that there are multiple receptors capable of binding Aβ assemblies in a physiological context and mediating their synaptotoxic actions. Although interaction of Aβ with each of the receptors has previously been shown using in vitro or cellular binding assays, the molecular details of the binding reaction remain unclear.

Recently, PrP$^C$ was identified as a high-affinity receptor for Aβo[10] an observation subsequently confirmed by other groups[19–23]. Binding of Aβo and PrP$^C$ has been demonstrated using both cellular and biochemical methods[10,19–23]. Binding of Aβo to PrP$^C$ has also been reported to initiate synaptotoxic signaling, causing suppression of long-term potentiation (LTP) and retraction of dendritic spines[10,24,25]. It has been proposed that this signaling pathway requires interactions between PrP$^C$ and mGluR5, resulting in activation of intracellular fyn kinase, with subsequent phosphorylation and redistribution of NMDA receptors[25–28]. In vivo, genetic deletion of PrP$^C$ rescues behavioral deficits, synaptic loss, and early mortality in AD transgenic models[29,30]. Compounds that block Aβ binding to PrP$^C$, or that inhibit activation of downstream signaling mechanisms, have been shown to ameliorate pathology in AD transgenic models[31,32], and some of these are being tested in human patients[33].

Amyloid fibril formation from soluble Aβ monomers is a well-characterized process that involves distinct, kinetically defined steps of primary nucleation, secondary nucleation, and elongation[34–36]. Several classes of molecules, including chaperones, antibodies, and small molecules, have been reported to influence specific steps of the polymerization process[35,37–40]. Our laboratory has recently investigated the effect of PrP$^C$ on Aβ polymerization[22]. That study, which relied upon biochemical assays and mathematical modeling, demonstrated that PrP$^C$ specifically inhibits the elongation step of Aβ polymerization, most likely by binding to the ends of growing fibrils. However, we

did not directly prove this mechanism by measurement of fibril elongation rates, or by localization PrP on individual fibrils.

In this study, we have employed super-resolution microscopy (SRM) to directly visualize, at a nanoscale level, the dynamics of Aβ assembly, and the interactions of PrP$^C$, as well as two other cell-surface receptors, with Aβ fibrils and neurotoxic oligomers. Analyzing several Aβ-receptor systems in parallel enabled us to reveal common molecular mechanisms by which these receptors interact with pathologically relevant Aβ aggregates to transduce neurotoxic signals. Altogether, our data provide insights into the molecular origins of AD, and they lay the groundwork for development of therapeutic approaches to block receptor-mediated Aβ neurotoxicity.

## Results
In this study, we have used direct stochastic optical reconstruction microscopy (dSTORM)[41] and structured illumination microscopy (SIM)[42] to visualize directly the effect of PrP and two other putative Aβ receptors on the process of Aβ polymerization with a resolution of 20 nm (dSTORM) or 100 nm (SIM) (Supplementary Fig. 1). At these resolutions, we were able to visualize Aβ oligomers, protofibrils and fibrils, and determine the localization of PrP on these structures. We polymerized synthetic Aβ1–42 labeled with either Cy3 or Cy5 under carefully defined conditions, which have been shown to result in reproducible kinetic curves as monitored by thioflavin T (ThT) fluorescence[34,35,43]. For colocalization experiments, recombinant PrP was fluorescently labeled by substituting a cysteine residue at position 34, and reacting it with a maleimide derivative of either Alexa Fluor 555 (AF555) or Alexa Fluor 488 (AF488). We first monitored the kinetics of fibril formation by ThT fluorescence to confirm that incorporation of the fluorescent labels into either Aβ or PrP did not affect the Aβ polymerization process or the ability of PrP to inhibit polymerization (Supplementary Fig. 2). As we reported previously[22], PrP in sub-stoichiometric amounts dramatically inhibited Aβ polymerization.

**PrP promotes formation of shorter, more numerous fibrils.** Based on its effect on the kinetics of Aβ polymerization, we previously concluded that PrP specifically inhibited the elongation step of polymerization[22]. In this case, it would be predicted that Aβ fibrils formed in the presence of PrP would be shorter than those formed in the absence of PrP. Moreover, the total number of fibrils would be increased in the presence of PrP, since the flux of monomers would be shifted from elongation toward nucleation events that generate additional fibrils[38]. To test these predictions, Aβ-Cy5 monomers at a concentration of 20 μM were polymerized for 24 h in the presence of different concentrations of PrP. The fibrils that formed were then imaged by SIM.

We found that PrP over a concentration range of 0.1–1 μM caused a dose-dependent reduction in the length of Aβ fibrils, and significantly increased the number of fibrils (Fig. 1a–d). To quantify these effects, we measured the length and number of all the fibrils resolved in each SIM image. We found that, when the PrP concentration was increased from 0 to 1 μM, the mean value of fibril length decreased progressively from $0.90 \pm 0.02$ μm to $0.29 \pm 0.01$ μm (Fig. 1e), while the number of fibrils increased from $0.13 \pm 0.005/\mu m^2$ to $0.66 \pm 0.02/\mu m^2$ (Fig. 1g). Although these preparations were heterogeneous, the size distribution of fibrils formed in presence of PrP was significantly different from control samples, with a preponderance of smaller species in the former samples (Fig. 1f, compare red curve with gray curves). In the absence of PrP, some fibrils reached lengths of up to 6 μm, and ~10% of the fibrils were >2 μm (Fig. 1f, inset). In contrast, even at lowest concentration of PrP (0.1 μM), no fibrils were

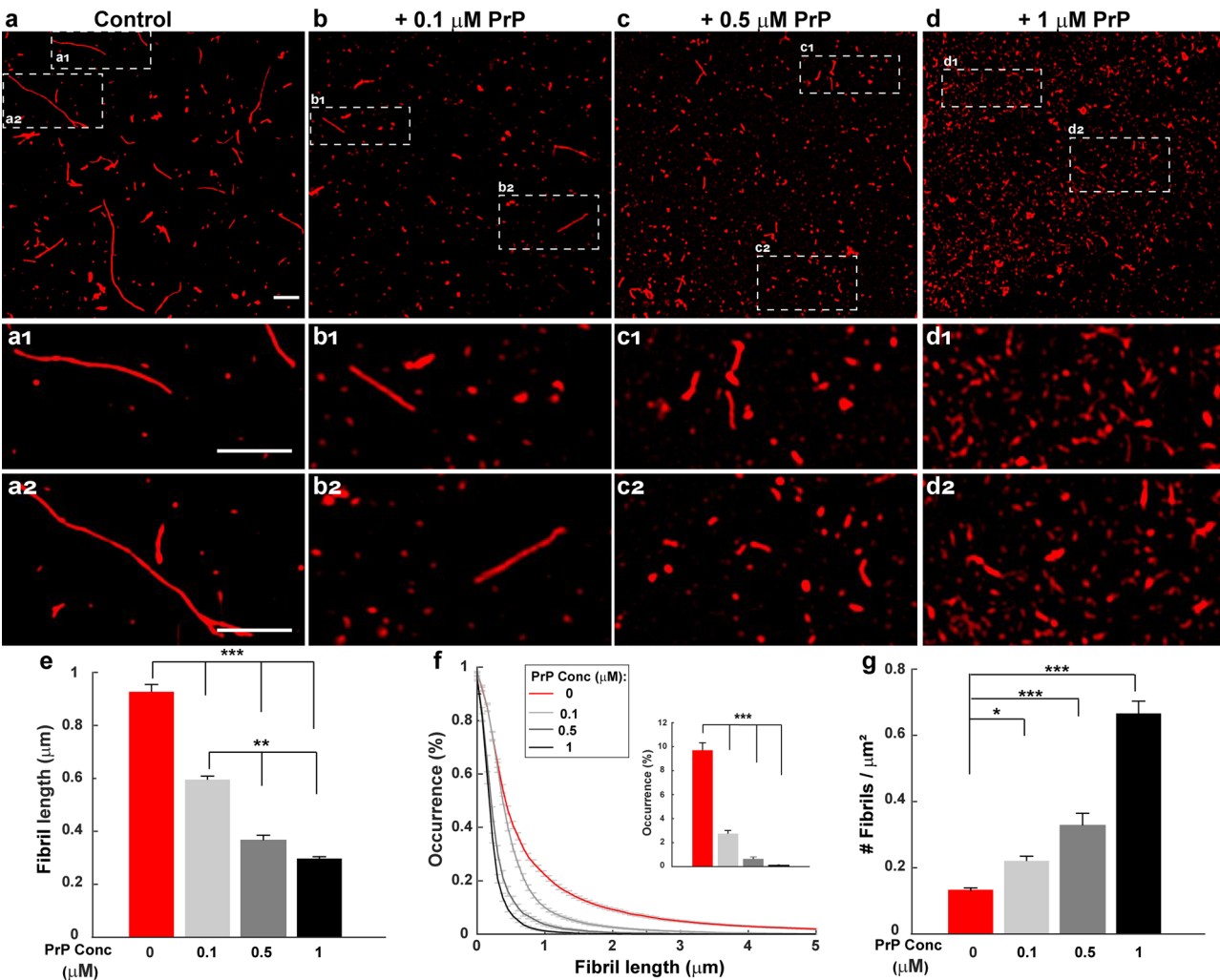

**Fig. 1 PrP promotes formation of shorter, more numerous Aβ fibrils.** Aβ-Cy5 monomer (20 μM) was polymerized for 24 h in the presence of 0 μM (**a**), 0.1 μM (**b**), 0.5 μM (**c**), or 1 μM (**d**) PrP-AF555. Fibrils were then imaged by SIM. Panels **a**$_{1,2}$–**d**$_{1,2}$ show boxed areas in (**a–d**), respectively, at higher magnification. Scale bars are 1 μm. **e** Bars show mean fibril length at each PrP concentration. **f** Cumulative distributions of fibril length at each PrP concentration. Inset indicates the number of fibrils larger than 2 μm. **g** Bars indicate the number of detectable Aβ-Cy5 fibrils/μm² at each PrP concentration. Data represent mean ± S.E. *P < 0.05, **P < 0.01 and ***P < 0.001 (two-sided Student's t-test). The numbers of biological independent samples (N) and analyzed images (X) in each conditions are: Control: N = 9, X = 44; +0.1 μM PrP: N = 6, X = 22; +0.5 μM PrP: N = 10, X = 20; +1 μM PrP: N = 5, X = 15.

longer than 3 μm, and only 2.7% were >2 μm. Thus, increasing the concentration of PrP shifted the population of Aβ fibrils to smaller sizes.

Taken together, these data suggest that PrP affects Aβ polymerization in a dose-dependent manner, resulting in the formation of shorter but more numerous fibrils at 24 h.

**PrP slows the growth of Aβ fibrils.** To characterize directly the effect of PrP on the dynamics of the elongation process, we measured the effect of PrP on fibril lengths at different points of the Aβ polymerization reaction. Each assay began with 20 μM Aβ in monomeric form. In the absence of PrP, Aβ monomers rapidly polymerized, reaching a maximum size distribution by 24 h, after which mean fibril size remained constant for up to seven days (Fig. 2a; quantitation in Fig. 2c, d). In contrast, the polymerization rate was slowed significantly when 0.5 μM PrP-AF555 was added to the reaction at the starting point. Under these conditions, the fibrils continued to grow slowly over seven days (Fig. 2b; quantitation in Fig. 2c, d). At each of the time points analyzed, the mean length of fibrils formed in the presence of PrP

was significantly less than in control conditions. In addition, the number of fibrils was higher in the presence of PrP at each time point, from 4 h to 7 days, after the initiation of polymerization (Fig. 2e). Taken together, these results indicate that PrP significantly slows the process of fibril elongation, and increases the number of fibrils formed.

**Aβ polymerization is strongly polarized, and PrP selectively blocks elongation at the more rapidly growing end.** It has been shown that elongation of Aβ fibrils is strongly polarized, with the two ends of the fibril growing at different rates (fast and slow)[44]. By adapting several published seeding procedures[45–47], we were able to determine how PrP affected the elongation rate at each end of the fibril. Sheared, preformed fibrils (referred to as seeds) labeled with Cy5 were allowed to grow by incubation in a solution of 10 μM monomeric Aβ labeled with Cy3 (Fig. 3a). After different lengths of time, ranging from 2 h to 7 days, the fibrils were imaged by SIM to visualize growth of the seeds at their ends. The lengths of the green extensions at the two ends of each red seed were measured over time to monitor the progress of elongation.

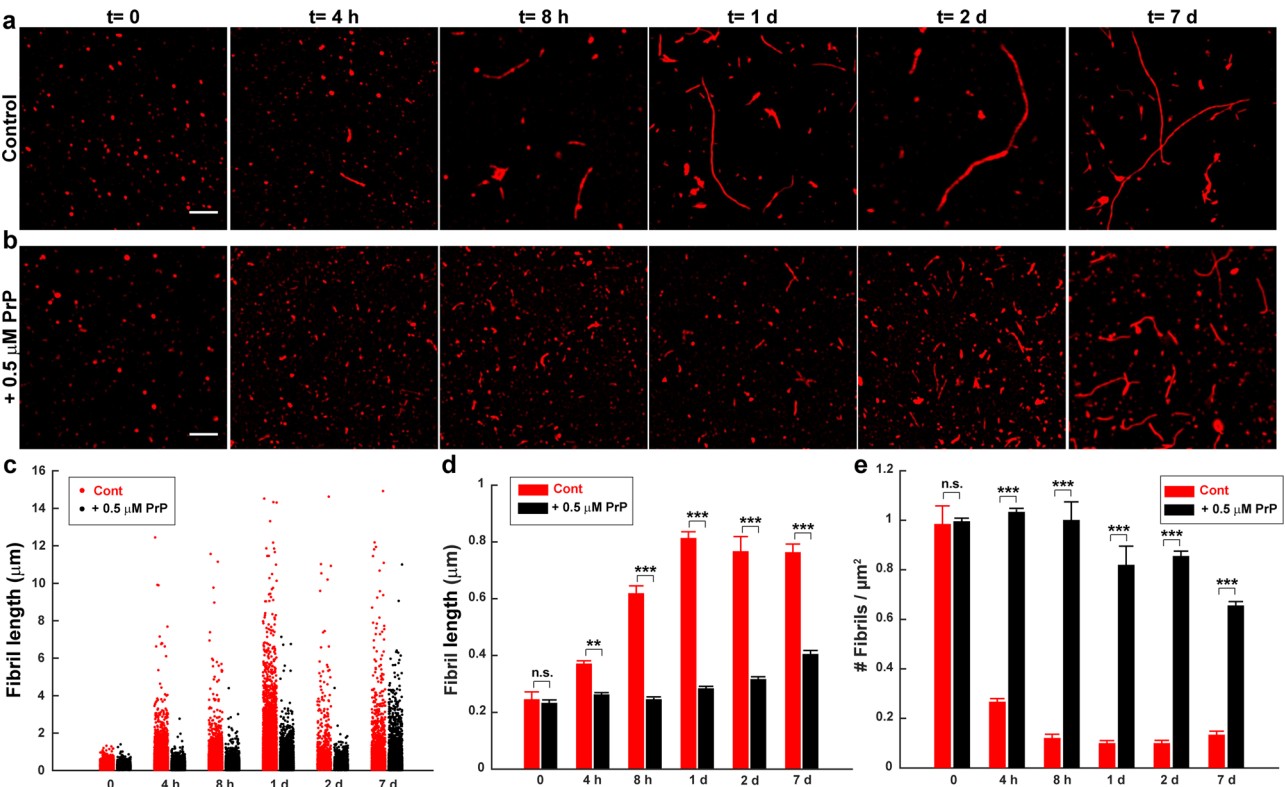

**Fig. 2 PrP slows the growth of Aβ fibrils.** Aβ-Cy5 monomer (20 μM) was polymerized for the indicated times in the presence of 0 μM (**a**) or 0.5 μM (**b**) PrP-AF555. Fibrils were then imaged by SIM. Scale bar in (**a**, **b**) (*t* = 0) is 1 μm. **c** Distributions of fibril lengths at each time point in the presence of 0 μm (red dots) or 0.5 μM PrP-AF555 (black dots). Each dot represents an individual fibril. **d**, **e** Bars indicate the mean length of fibrils (**d**) and the number of detectable Aβ-Cy5 fibrils/μm² (**e**) at each time point for 0 and 0.5 μM PrP-AF555. Data represent mean ± S.E. *P < 0.05, **P < 0.01 and ***P < 0.001 (two-sided Student's *t*-test). The numbers of biological independent samples (N) and analyzed images (X) in each conditions are: N ≥ 3, X ≥ 6.

Using this assay, we confirmed a previous report[44] that fibril growth under control conditions occurs asymmetrically at the two ends (Fig. 3a and b). At each time point, one end of every fibril had grown more than the other end, as revealed by a scatterplot of the lengths of the green extensions at the two ends of 390 seeds (Fig. 3e). There was no statistical correlation between the lengths of the extensions at the two ends of each fibril, indicating that the two ends behaved independently. We noted that some seeds grew only at one end during the course of the experiment, a phenomenon that was observed previously, and was attributed to longer paused periods in fibril growth at the slow end[44]. We have designated the longer (faster-growing) extension as End 1, and the shorter (slower-growing) extension as End 2. At early time points (2 h), the mean lengths of the fast- and slow-growing ends were 0.50 ± 0.13 and 0.11 ± 0.02 μm, respectively (Fig. 3g and i). By 24 h, the mean lengths of the fast and slow-growing ends had reached 2.09 ± 0.13 μm and 0.32 ± 0.03 μm, respectively, after which they changed relatively little for up to seven days (Fig. 3g and i).

When 0.5 μM PrP was added to the reaction together with fresh monomers, the growth characteristics were markedly different (Fig. 3c and d). Green extensions were seen at only one end of most fibrils (Fig. 3f), and these extensions grew slowly, with mean lengths of 0.14 ± 0.017 at 2 h, 0.34 ± 0.017 at 1 day, and 0.94 ± 0.08 at 7 days (Fig. 3h and i). The elongation values at all but the 7-day time point were statistically indistinguishable from those measured at the slow-growing end in the absence of PrP (Fig. 3i, compare black and red dashed lines). Taken together, these data suggest that PrP completely blocks elongation at the fast-growing end of the fibril, with growth of the fibril restricted to elongation at the slow-growing end. However, because the data shown in Fig. 3

represent ensemble measurements of a population of fibrils, we cannot formally rule out the possibility that PrP is simultaneously altering growth rates at both ends of the fibril. We regard this possibility as unlikely, however, based on the next set of experiments, in which we directly visualized the location of PrP on individual Aβ fibrils.

**PrP binds exclusively to the fast-growing end of Aβ fibrils.** A more likely mechanism by which PrP blocks fibril elongation is by binding selectively to the fast-growing end of the fibril, preventing further monomer addition at that end, without any effect on elongation at the slow-growing end. To directly localize PrP on individual fibrils, we polymerized Aβ-Cy5 monomers in the presence of different concentrations of PrP-AF555 for 24 h, and then visualized Aβ aggregates by SRM. Strikingly, dual-color dSTORM images showed that PrP-AF555 was selectively associated with only one end of each Aβ fibril (Fig. 4a). As would be expected, the number of aggregates with co-localized PrP increased as the concentration of PrP was raised (Fig. 4b).

To demonstrate that the observed association of PrP with Aβ aggregates was specific, and not the result of random colocalization, we employed a randomization test. First, the position of PrP in each image was digitally shifted in a random direction along a two-dimensional vector of defined size, and then the colocalization index was recomputed. We performed this analysis for different vector sizes ranging from 100 nm to 4 μm (Fig. 4c). We found that the colocalization index decreased rapidly as the size of the vector was increased, reaching a minimum level, corresponding to random placement. As a control, we measured the colocalization between PrP and Aβ in unrelated images from two different experiments (red curve in Fig. 4c). In this case, very low

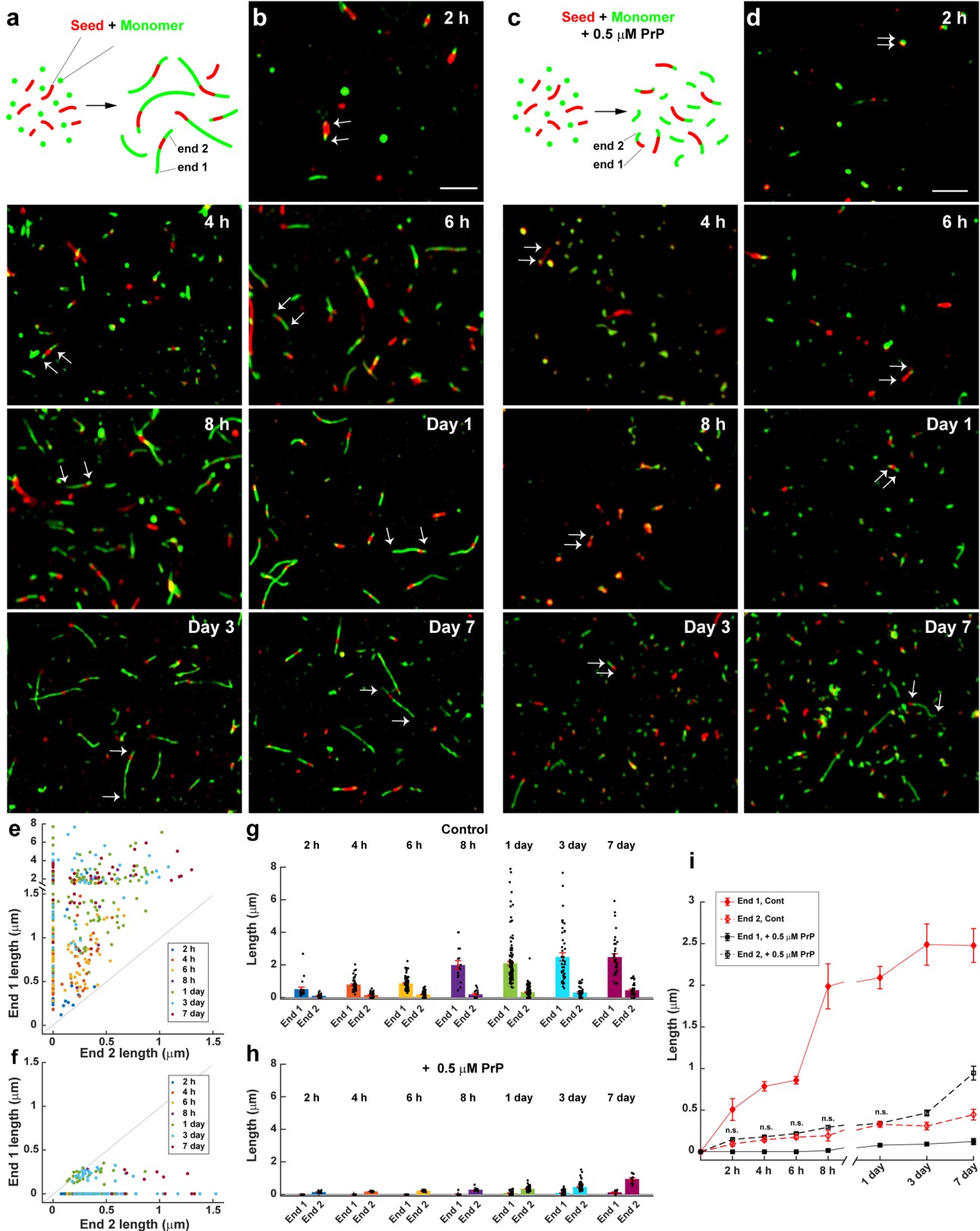

colocalization was measured, and this value remained constant when the size of the vector increased. Taken together, these results demonstrate that PrP associates non-randomly with individual Aβ aggregates.

We next adopted an unbiased statistical method to quantify the localization of PrP with respect to fibril ends for large populations of individual Aβ fibrils over a range of different PrP concentrations. We measured the minimum distance ($D_{min}$) between each fluorescent PrP spot and the closest end of the associated fibril, and then normalized this distance to the total length ($L_{total}$) of the fibril to give the quantity $D_{min}/L_{total}$ (see cartoon in Fig. 4d). $D_{min}/L_{total}$ values could, theoretically, vary between 0 (PrP exactly at

**Fig. 3 Aβ polymerization is strongly polarized, and PrP selectively blocks elongation at the fast-growing end. a** Schematic representation of a seeding assay, in which fresh monomers labeled with Cy3 (green) were added to sheared, preformed fibrils (seeds) labeled with Cy5 (red). The two ends of each seed elongate at different rates, resulting in long and short green extensions, designated End 1 and End 2, respectively. **b** Two-color SIM images acquired at the indicated times after addition of Aβ-Cy3 monomers (10 μM) to Aβ-Cy5 seeds (10 μM monomer equivalent). Arrows in each panel indicate the elongation of the seed at the two ends. Scale bar in (**b**) (2 h) is 2 μm. **c** Schematic representation of the seeding assay in the presence of 0.5 μM PrP. **d** Two-color SIM images acquired as in (**b**), but in the presence of 0.5 μM PrP. Arrows in each panel show that seeds elongate at only one end. Scale bar in (**d**) (2 h) is 2 μm. **e, f** Scatterplots showing the lengths of End 1 and End 2 over time for each detected seed in the absence of PrP (**e**) and in the presence of 0.5 μM PrP (**f**). The total number of seeds measured was 390 and 215 for 0 μM and 0.5 μM PrP, respectively. **g, h** Bars indicate the mean lengths of End 1 and End 2 over time in the absence of PrP (**g**) and in the presence of 0.5 μM PrP (**h**). Data represent mean ± S.E. **i** Change in the mean lengths of End 1 and End 2 over time, with and without PrP. These are the same data as in (**g, h**), but plotted to allow easier comparison of the different conditions. n.s., no statistically significant difference between End 2 length with and without PrP at the indicated time points. Error bars represent mean ± S.E. *$P < 0.05$, **$P < 0.01$ and ***$P < 0.001$ (two-sided Student's $t$-test). Numbers of biological independent samples ($N$) and analyzed images ($X$) in each time point are: $N \geq 3$, $X \geq 9$.

the fibril end) and 0.5 (PrP at the mid-point of the fibril). We chose this normalization procedure, since fibril length decreases significantly with increasing amounts of PrP (Figs. 1 and 2). When we plotted the distribution of the $D_{min}/L_{total}$ values for a large number of fibrils, we found that, at each of three different PrP concentrations, the majority of $D_{min}/L_{total}$ values were less than 0.1, indicating that the PrP spot was localized very close to one end of the underlying fibril (Fig. 4d–f). At 0.1, 0.5, and 1 μM PrP, the proportion of $D_{min}/L_{total}$ values <0.1 was 77%, 75%, and 68%, respectively. Thus, even for the very short fibrils present at high PrP concentrations, the PrP spot was localized asymmetrically, closer to one end of the fibril. At all three tested concentrations, the localization of PrP on Aβ fibrils was significantly different from a random distribution, which was determined by measuring $D_{min}/L_{total}$ distances in images from unrelated PrP and Aβ experiments (Fig. 4g). In this case, only 30% of the $D_{min}/L_{total}$ values were less than 0.1.

We next wished to determine whether the fibril end where PrP bound was the fast- or slow-growing end. To address this question, we took advantage of our seeding assay, in which we could clearly resolve fast and slow-growing ends of individual fibrils (Fig. 4h). In these experiments, fibrils were polymerized by addition of 10 μM Aβ-Cy3 monomer (green) to short, preformed seeds consisting of Aβ-Cy5 (red). After 24 h of incubation at 37 °C, PrP-AF488 (magenta) was added to seeded fibrils, and samples were then imaged by three-colour SIM. We found that PrP-AF488 was selectively associated with the end of the fibril with the longer green extension, indicating binding to the fast-growing end of the fibril (Fig. 4i and Supplementary Fig. 3). This localization was consistent over a range of different lengths of the extensions from the fast-growing end, and was not correlated with the lengths of the extensions from the slow-growing end (Fig. 4j). These data indicate that PrP inhibits fibril elongation by binding selectively to the fast-growing end of the fibril, thereby blocking growth at that end. The localization of PrP exclusively at the fast-growing end of the fibril makes it unlikely that PrP also affects fibril elongation at the slow-growing end, which can lie many microns away, depending on the length of the fibril. Real-time growth experiments on individual Aβ fibrils using SRM will allow definitive resolution of this question.

Finally, we wished to investigate the stoichiometry of PrP binding to fibrils ends. In order to determine whether each fibril end bound one or more molecules of PrP, we performed a triple-label experiment (Supplementary Fig. 4) in which Aβ-Cy5 monomers were incubated for 24 h in the presence of an equimolar (50:50) mixture of PrP labeled with either Alexa Fluor 488 (PrP-AF488) or Alexa Fluor 555 (PrP-AF555). Samples were then imaged with three-colour SIM. In each image, we measured the number of Aβ-associated PrP clusters containing either AF488 or AF555, as well as the number of PrP clusters containing

both fluorophores. We determined that 58 ± 0.03% of the fibril-associated PrP clusters were labeled with both AF488 and AF555, indicating that the majority of clusters contained more than one molecule of bound PrP. Several such clusters are indicated by arrows in Supplementary Fig. 4a. These data indicate that most fibril ends bind more than one PrP molecule. We also found that only 25 ± 0.03% of the PrP clusters not associated with fibrils contained both labels, indicating that most unbound PrP molecules were monomeric. This result argues against artifactual aggregation of PrP during the course of the experiment.

**Localization of PrP on neurotoxic Aβ assemblies.** Our results thus far have focused on the localization of PrP on Aβ fibrils. However, small oligomeric assemblies of Aβ, rather than long fibrils, are thought to be the most neurotoxic species, and to be primarily responsible for synaptic loss and neurological dysfunction in AD[5–7]. We, therefore, sought to define the localization of PrP on two kinds of neurotoxic Aβ assemblies: Aβ-derived diffusible ligands (ADDLs), which are oligomeric structures[48]; and protofibrils, which are short, fibril-like structures[49]. These two kinds of Aβ assemblies, although differing in their method of preparation and structural properties, share the common feature of being highly neurotoxic when tested in cellular, brain slice, and in vivo assays.

We used a standard and widely used method[48,50] to produce ADDLs and protofibrils, based on resuspending a dried film of Aβ peptide in tissue culture medium, and incubating it for different periods of time. After 16 h of incubation the sample contains mostly small, oligomeric assemblies (ADDLs), while incubation for 3 days and 7 days results in increased numbers of protofibrils. First, we used EM to characterize the structures of the ADDL and protofibril preparations used in our experiments. (Supplementary Fig. 5). Consistent with many previously published studies[48], ADDLs consisted primarily of a heterogeneous population of globular and ellipsoid structures with the majority having a diameter of 5–15 nm, just at the resolution limit of dSTORM. In protofibril preparations formed by incubation of ADDLs for 3 days, short, worm-like assemblies with a mean length of 16.0 nm began to appear. After 7 days of incubation, these assemblies became longer, with a mean size of 49.2 nm. In contrast, mature fibrils polymerized directly from monomeric Aβ for 24 h were much longer (see Figs. 1 and 2). The worm-like structures observed in the two protofibril preparations displayed an irregular surface, in contrast to the smooth surface of mature fibrils. The morphology of the ADDL preparations incubated for 3 and 7 days is consistent with published images of protofibrils[21].

We next set out to localize PrP on ADDLs and Aβ protofibrils. In the first set of experiments, ADDLs, protofibrils and fibrils were prepared using Cy5-labeled peptide, and were then incubated with PrP-AF555. Samples were then imaged with dual-color dSTORM.

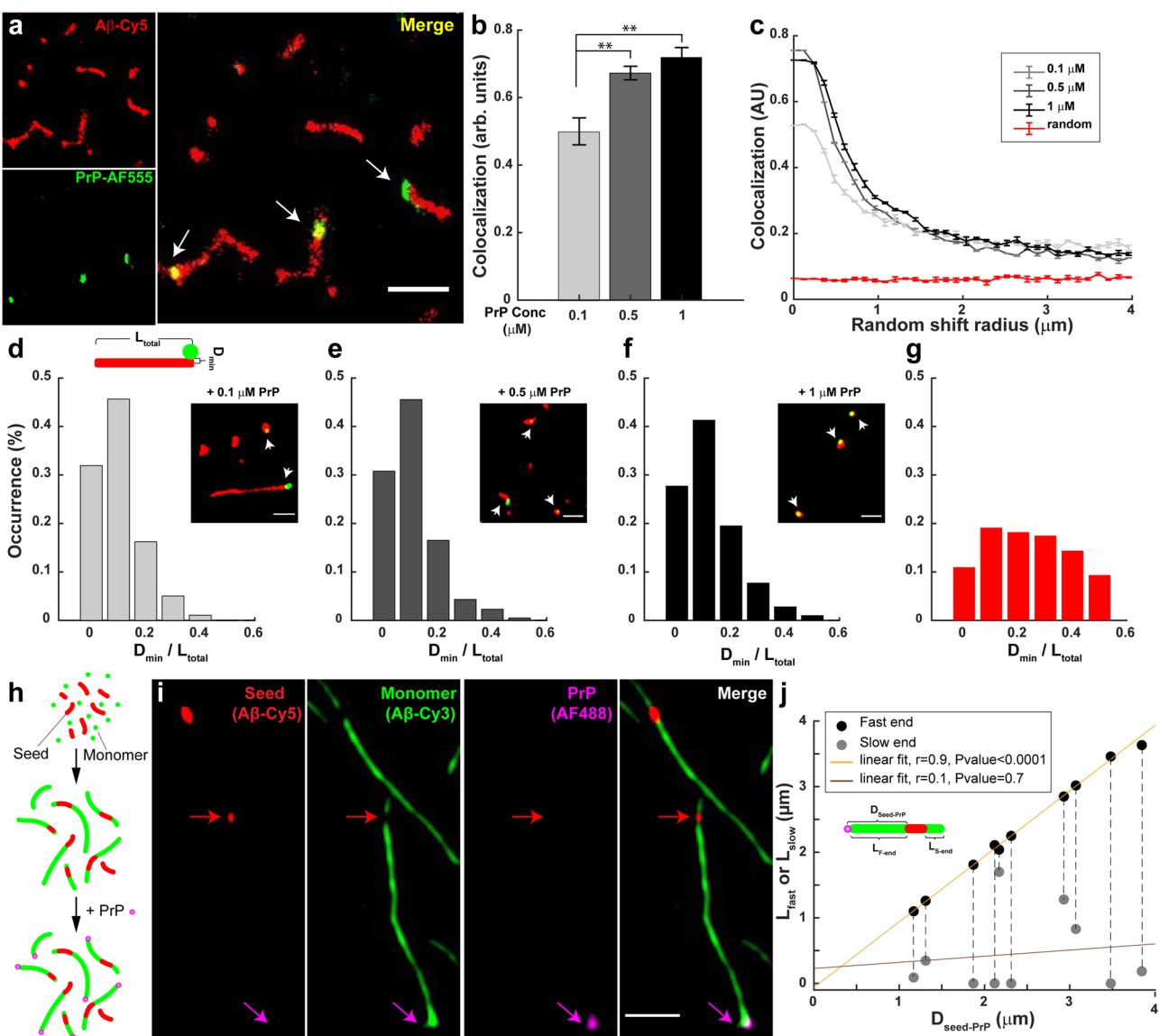

**Fig. 4 PrP binds exclusively to the fast-growing end of Aβ fibrils. a** dSTORM images of Aβ-Cy5 (20 μM) polymerized for 24 h in the presence of 0.5 μM PrP-AF555. Arrows in the merged image indicate the localization of PrP near the ends of Aβ fibrils. Scale bar is 1 μm. **b** Bars indicate the colocalization between Aβ and PrP-AF555 at different concentration of PrP. Data represent mean ± S.E. **$P < 0.01$ and ***$P < 0.001$ (two-sided Student's $t$-test). **c** Colocalization between Aβ and PrP decays after a set of random direction shifts, and approaches the values derived from unrelated PrP and Aβ images from two different experiments (red line). Error bars represent mean ± S.E. **d–f** Aβ-Cy5 (20 μM) was polymerized for 24 h in the presence of either 0.1 μM, 0.5 μM, or 1 μM PrP-AF555, and samples were imaged by SRM. Histograms show the distribution of $D_{min}/L_{total}$ values (see cartoon in **d**) for 95-227 fibrils with associated PrP spots. Insets show the dSTORM images of Aβ fibrils (red) formed in the present of different concentration of PrP-AF555 (green). Scale bars are 0.5 μm. **g** Random distribution of $D_{min}/L_{total}$ values derived from unrelated PrP and Aβ images from two different experiments. The number of analyzed SIM images in +0.1 μM PrP, +0.5 μM PrP, +1 μM PrP and random condition are: $X = 14$; 10; 10, and 17, respectively. **h** Schematic representation of three-color imaging assay. **i** Individual fibrils **i**maged for Cy5 (Aβ seed), Cy3 (Aβ monomer), AF488 (PrP), and a merge of the three colors. The magenta arrow indicates PrP bound to the fast-growing end of a single fibril. The red arrow indicates the position of the seed. Scale bar is 1 μm. **j** Distance between the seed and the PrP spot ($D_{seed-PrP}$) plotted against the lengths of the fast- and slow-growing ends ($L_{fast}$ and $L_{slow}$, respectively) for 10 separate fibrils. The pairs of black and gray dots connected by a dotted line correspond to the two ends of each fibril. $r$ represent Pearson's correlation coefficient and corresponding $P$-value calculated using a two-sided Student's $t$ test.

As expected, PrP and ADDLs co-localized with a high degree of overlap (Fig. 5a). Interestingly, we often observed that PrP localized eccentrically on globular or ellipsoid-shaped ADDL aggregates, being closer to one edge of the aggregate (Fig. 5a₁). This effect was apparent when we calculated $D_{min}/L_{total}$ values for a large number of aggregates; these values were not evenly distributed, with 82% of the aggregates displaying $D_{min}/L_{total}$ values <0.1, indicative of an asymmetric localization (Fig. 5b). On protofibrils, which typically displayed identifiable ends, we observed that PrP was bound

exclusively to one end (Fig. 5c and c₁), similar to its localization on longer Aβ fibrils (Fig. 5e and e₁). On protofibrils and fibrils, 56 and 73% of the $D_{min}/L_{total}$ values, respectively, were <0.1 (Fig. 5d and f).

In a second set of experiments, we determined the effect of PrP on protofibril formation. PrP was added to ADDL preparations, and samples were then incubated for one week to allow the formation of protofibrils. We found that protofibrils formed in the presence of PrP were significantly shorter than species formed in control samples, and PrP again bound exclusively to one end of

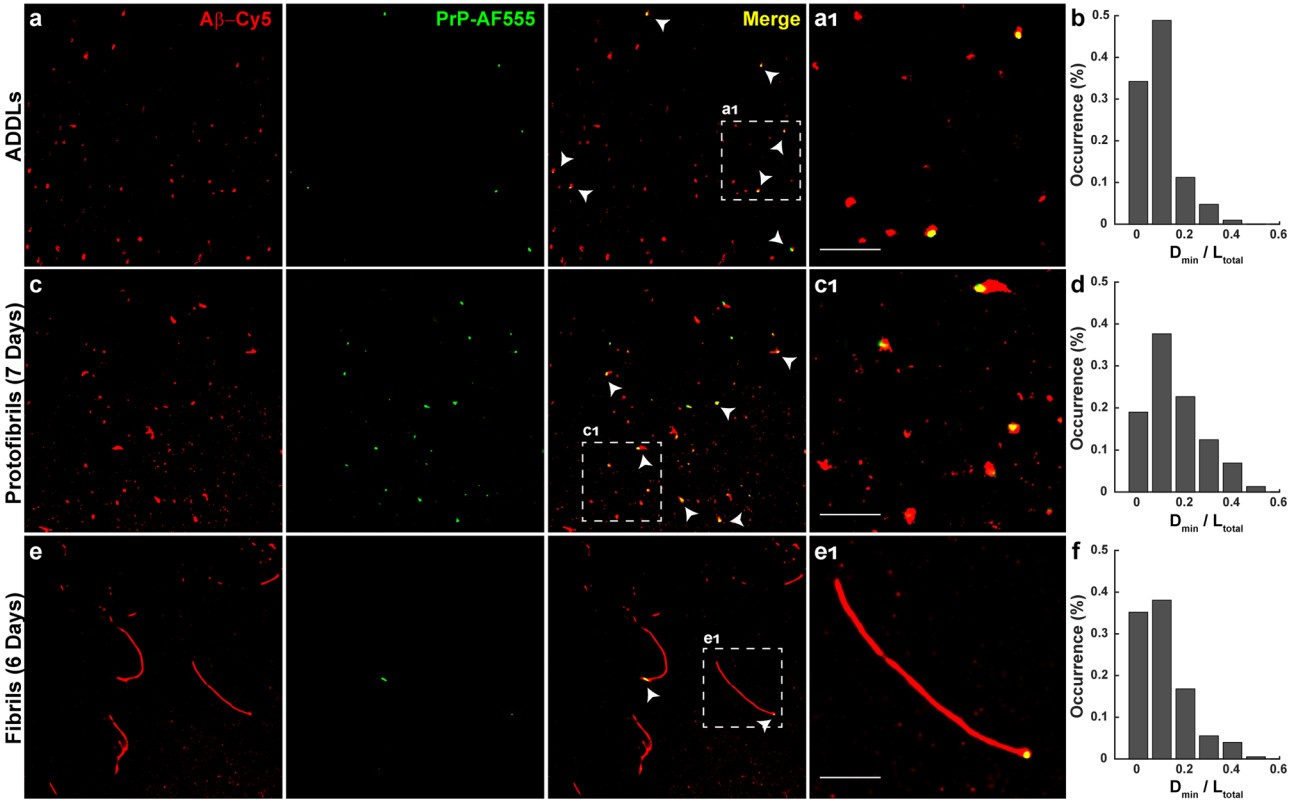

**Fig. 5 Localization of PrP on neurotoxic Aβ assemblies.** Pre-formed ADDLs (**a**), protofibrils (**c**), and fibrils (**e**) (all at 20 μM monomer-equivalent concentration) were incubated with 0.5 μM PrP-AF555 and then imaged by dSTORM. Panels **a₁**, **c₁**, and **e₁** show boxed areas in (**a, c, e**), respectively, at higher magnification. Scale bars are 1 μm. Histograms (**b, d, f**) show the distribution of $D_{min}/L_{total}$ values, calculated as in Fig. 4, for ADDLs, protofibrils, and fibrils, respectively. The number of analyzed SIM images in oligomer, protofibrils, and fibrils condition are: $X = 6$; 10 and 18, respectively.

each protofibril (Supplementary Fig. 6). Taken together, these data suggest that PrP interacts selectively with oligomers and protofibril ends in a fashion similar to its interaction with the ends of longer Aβ fibrils.

**Effect of PrP on neurotoxicity of Aβ assemblies.** Given the specific and spatially localized binding of PrP to Aβ oligomers and protofibrils, we asked whether this interaction played a role in the neurotoxic activity of these forms in a relevant biological system. We predicted that binding of exogenous, recombinant PrP to neurotoxic Aβ assemblies would block their ability to bind to PrP[C] on the neuronal surface, and thereby reduce their ability to induce neurotoxic effects. To test this prediction, we assayed the neurotoxicity of several kinds of Aβ aggregates by measuring their ability to induce retraction of dendritic spines on cultured hippocampal neurons[51]. We and others have demonstrated that Aβ oligomers cause rapid changes in the morphology and function of dendritic spines, and that these synaptotoxic effects depend on expression of PrP[C] by target neurons[25,52].

We found, first, that synaptotoxicity was correlated with the size of Aβ aggregate. ADDLs and 3-day protofibrils induced more spine retraction than 7-day protofibrils, (Supplementary Fig. 7a–f). These data confirm previous evidence that smaller oligomers are more toxic than fibrils[5], and they are consistent with the idea that smaller assemblies display a higher molar concentration of binding sites for cell-surface receptors like PrP[C] that transduce neurotoxic signals. We next incubated pre-formed ADDL and protofibril preparations with soluble, recombinant PrP prior to addition to neuronal cultures. We observed that pre-treatment with soluble PrP partially suppressed spine retraction

by these Aβ preparations, and this effect was dependent on the concentration of PrP added (Supplementary Fig. 7g–j).

We then performed a second kind of experiment to investigate how the toxicity of Aβ is affected by polymerization in the presence of PrP. Since inclusion of PrP in the polymerization reaction results in the accumulation of large numbers of short Aβ fibrils (Fig. 1), we asked whether these short fibrils were neurotoxic. In these experiment, we first polymerized Aβ monomers (5 μM) in the presence of increasing concentration of PrP (0, 0.1, 0.5, and 1 μM) for 24 h (Fig. 6a). The Aβ aggregates that formed were then added to cell media at a final concentration of 500 nM (Aβ monomer-equivalents) and incubated for 1 h. We found that Aβ samples polymerized with PrP were highly neurotoxic, and their toxicity increased as the concentration of PrP present in the reaction increased (Fig. 6b–d), correlating with the presence of shorter and more numerous fibrils (Fig. 1). In addition, we observed that the toxicity of these short fibrils was blocked by addition of excess recombinant PrP (5 μM) to the polymerization reaction, prior to dilution into tissue culture medium for treatment of the neurons (Fig. 6e, f). This rescuing effect was also observed when fibrils were polymerized using a higher concentration of Aβ (20 μM) (Supplementary Fig. 8). This experiment demonstrates that the presence of PrP during the polymerization reaction favors the accumulation of short, highly neurotoxic fibrils, but that saturation of these fibrils with additional PrP reduces their toxic effect.

Taken together, both of these experiments demonstrate that addition of exogenous, recombinant PrP to several different Aβ assemblies significantly reduces their neurotoxic effect on cultured hippocampal neurons. This result is consistent with the idea that the added PrP competes with cellular PrP[C] for

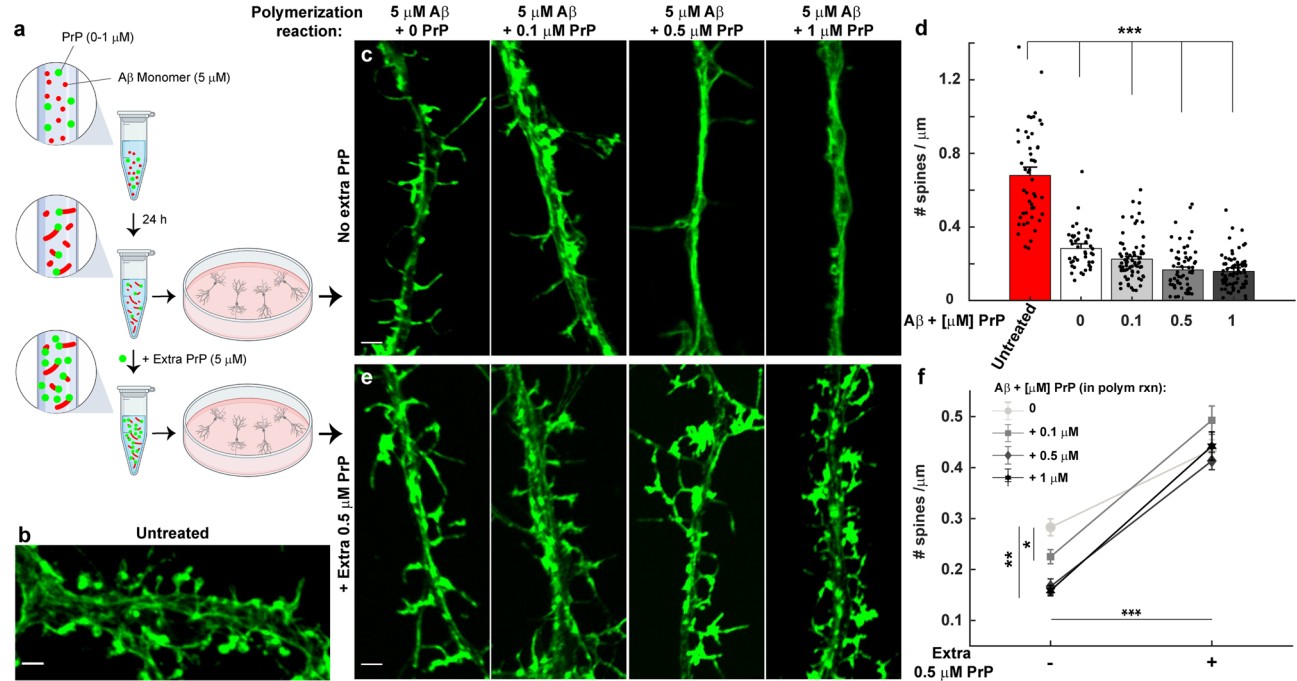

**Fig. 6 Neurotoxicity assay of Aβ fibrils polymerized in the presence of PrP. a** Schematic representation of a neurotoxicity assay, in which Aβ monomers (5 μM) were polymerized in the presence of PrP (0, 0.1, 0.5, and 1 μM) for 24 h. The resulting fibrils were then either diluted ten-fold directly into neuronal culture medium to give a final Aβ concentration of 500 nM (monomer-equivalents); or they were first incubated for 10 min with additional recombinant PrP to give a total concentration of 5 μM (equimolar to Aβ) before dilution into the culture medium (final PrP concentration of 0.5 μM). Neurons were fixed after 1 h of treatment and stained with Alexa 488-labeled phalloidin to visualize dendritic spines. The schematic was created with BioRender.com. **b** Confocal image of untreated hippocampal neurons, showing many normal, mushroom-shaped dendritic spines. **c** Neurons treated with Aβ fibrils polymerized in the presence of 0, 0.1, 0.5, and 1 μM PrP. Inclusion of PrP in the polymerization reaction increases the toxicity of the resulting fibrils, reflected in retraction of dendritic spines. Scale bars 2 μm. **d** Quantitation of spine number per μm for neurons treated as in (**c**). Data are presented as mean ± S.E. ***$P < 0.001$ (two-sided Student's $t$-test). The numbers of biological independent samples in each condition is ≥2 and the number of analyzed neurites in control, - PrP, +0.1 μM PrP, +0.5 μM PrP and +1 μM PrP condition are: $X = 48, 42, 72, 61,$ and 76, respectively. **e** Aβ polymerized in the presence of PrP, as in (**c, d**), were incubated with extra recombinant PrP (5 μM) before dilution into the neuronal culture medium. Incubation with excess PrP at the end of the polymerization reaction blocks the toxicity of fibrils. **f** Quantitation of spine number per μm with and without addition of excess recombinant PrP. Data are presented as mean ± S.E. *$P < 0.05$, **$P < 0.01$, and ***$P < 0.001$ (two-sided Student's $t$-test). This experiments repeated twice and the number of analyzed neurites in - PrP, +0.1 μM PrP, +0.5 μM PrP and +1 μM PrP condition are: $X = 59, 44, 55,$ and 53, respectively.

binding to these Aβ species, and, therefore, supports the role of endogenous $PrP^C$ as a receptor that mediates Aβ neurotoxicity.

**Other putative receptors interact with Aβ in a manner similar to $PrP^C$.** Having established a model for Aβ-$PrP^C$ interactions, we asked whether other putative receptor proteins interacted with Aβ aggregates via a similar mechanism. We decided to focus on FcγRIIb and LilrB2, since there is strong evidence that these receptors bind Aβ oligomers, and transduce neurotoxic signals in biological assays[11,12,53]. FcγRIIb, which is expressed in B cells, macrophages, neutrophils, as well as neurons, binds antigen-bound IgG complexes and transduces an inhibitory signal that results in inhibition of the B-cell-mediated immune response[54]. LilrB2 (whose mouse ortholog is called PirB) was originally thought to function exclusively in the immune system, but is now known to be expressed by neurons and to be involved in neurodevelopmental events[55].

For these experiments, we used the purified, extracellular domains of FcγRIIb and LilrB2, which encompass the Ig domains known to represent the Aβ binding sites[11,12]. First, we analyzed the effect of each of these receptor proteins on Aβ polymerization kinetics using ThT fluorescence (Fig. 7). We found that both FcγRIIb and LilrB2 inhibited Aβ polymerization in a manner very similar to $PrP^C$ (Fig. 7a–c), while a control protein, calmodulin, had no effect (Fig. 7d). In sub-stoichiometric amounts, each of

these receptor proteins increased the half-time required to reach a plateau value of ThT fluorescence (Fig. 7e). In contrast, the half-time remained constant for even very high concentrations of calmodulin.

We also analyzed the effect of these receptor proteins on Aβ fibril formation by SRM (Fig. 8). Strikingly, we found that both FcγRIIb and LilrB2 affect Aβ fibril length and number in a manner very similar to PrP. Thus, Aβ aggregates formed in the presence of FcγRIIb and LilrB2 were shorter and more numerous than under control conditions (Fig. 8g–l and m–r, respectively). As is the case for PrP, these effects were concentration-dependent, and occurred with substoichiometric levels of the receptor proteins. In contrast, fibrils formed in the presence of different concentrations of calmodulin, were very similar to control fibrils in terms of their length and number (Fig. 8a–f).

## Discussion

There has been considerable interest in identifying the cell-surface receptors that mediate the neurotoxic effects of Aβ oligomers, in part because of the possibility that small molecules targeting these receptors, or the downstream pathways they activate, could be used as therapeutic agents to treat AD. At least 10 different cell-surface proteins have been proposed to act as Aβ receptors[8,9]. However, there has been controversy about the functional relevance of these receptors, and how much each

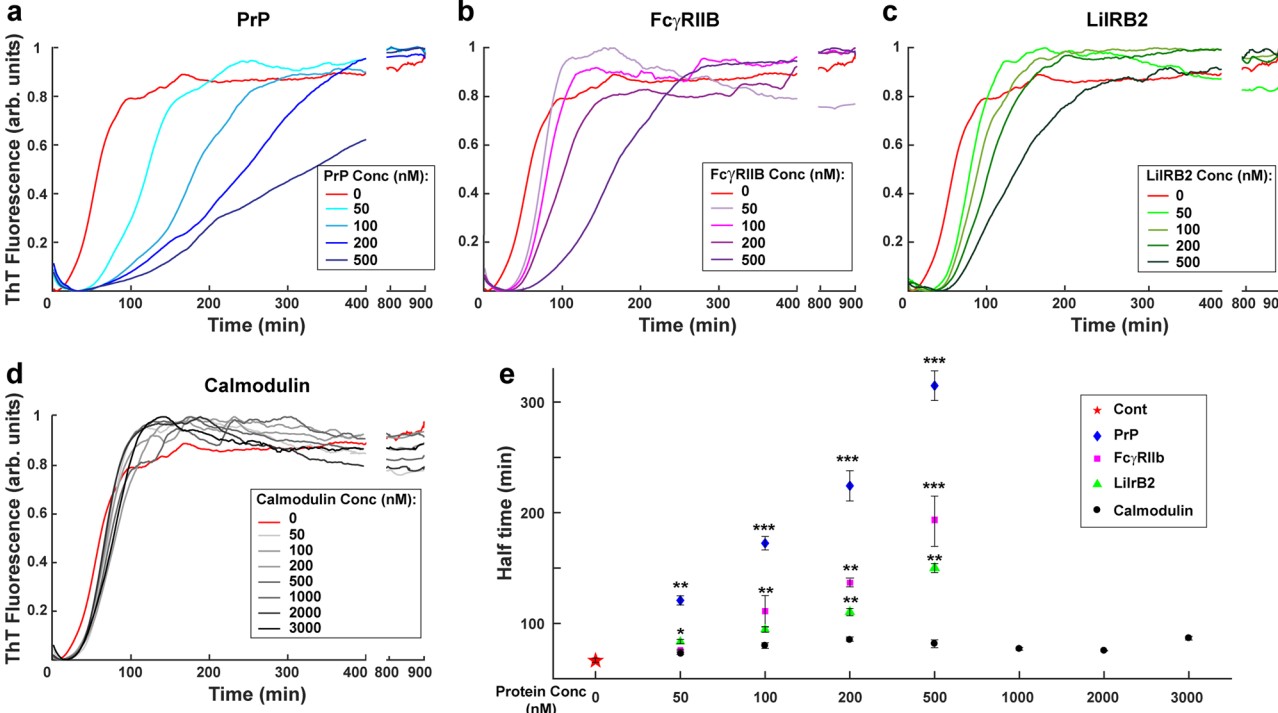

**Fig. 7 FcγRIIb and LilrB2 affect the kinetics of Aβ polymerization in a manner similar to PrP.** ThT curves for polymerization of unlabeled Aβ (5 μM) in the presence of increasing concentrations of recombinant PrP (**a**), FcγRIIb (**b**), LilrB2 (**c**) and calmodulin (**d**). **e** Effect of receptors on the half-times for Aβ polymerization, derived from the data in (**a–d**). Data represent mean ± S.E. Half-time values that are significantly different from control are indicated: *$P <$ 0.05, **$P <$ 0.01, and ***$P <$ 0.001 (two-sided Student's t-test). Each curve shown in **a–d** is the average of 3 replicates, and each condition repeated at least 5 times.

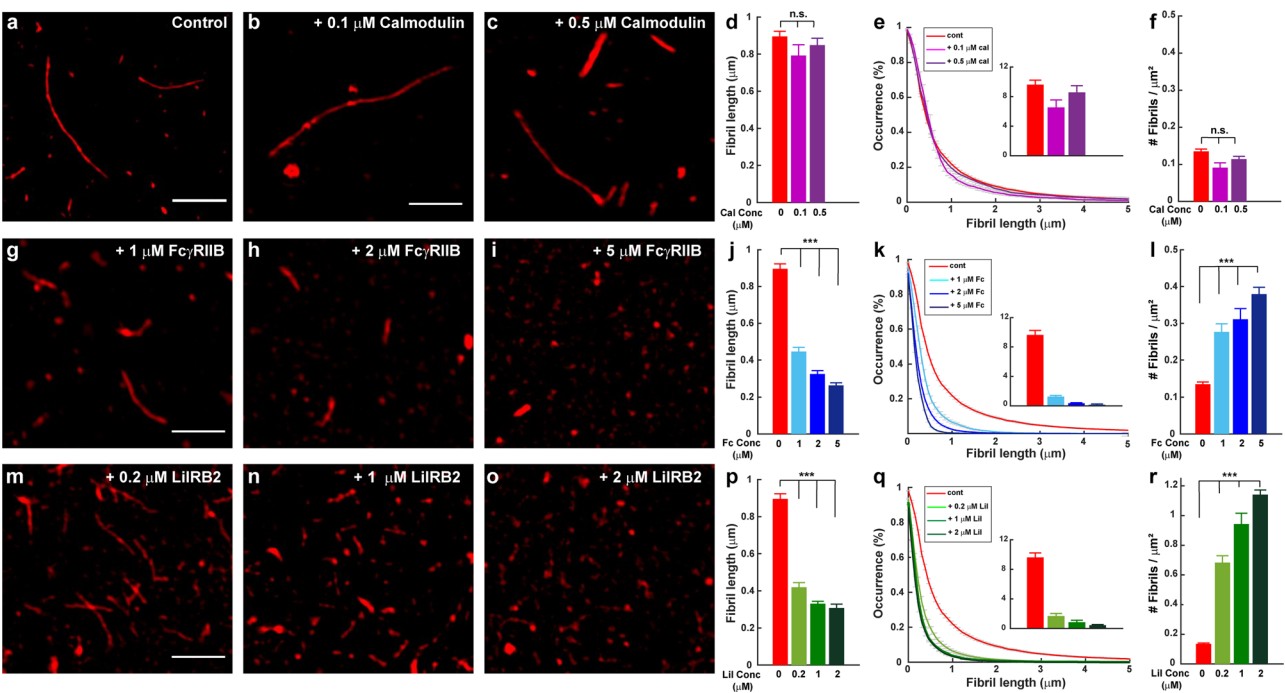

**Fig. 8 FcγRIIb and LilrB2 promote formation of shorter, more numerous Aβ fibrils.** Aβ-Cy5 monomer (20 μM) was polymerized for 24 h under control conditions, or in the presence of calmodulin (0.1 and 0.5 μM), FcγRIIb (1, 2, and 5 μM), or LilrB2 (0.2, 1, and 2 μM). **a–c**, **g–i**, and **m–o** show SIM images of the resulting fibrils. Scale bars are 2 μm. **d**, **j**, and **p** show mean fibril length under each condition. **e**, **k**, and **q** show cumulative distributions of fibril length; the insets indicate the number of fibrils larger than 2 μm. **f**, **l**, and **r** show the number of fibrils/μm². Data represent mean ± S.E. and ***$P <$ 0.001 (two-sided Student's t-test). n.s., not statistically significant. Numbers of biological independent samples (N) and analyzed images (X) in each condition are: $N > 3$ and $X > 10$, respectively.

contributes to the pathogenesis of AD. Part of the uncertainty on this subject derives from lack of detailed structural and molecular information about how these receptors interact with different kinds of Aβ aggregates. While many previous studies have characterized the binding reaction using biochemical or cell-based methods, we have employed SRM to directly visualize Aβ-receptor interactions at nanoscale resolution. This approach has allowed us to elucidate a unique structural mechanism by which one important receptor, PrP$^C$, influences the Aβ assembly process, and how it interacts with both Aβ fibrils as well as neurotoxic Aβ protofibrils and oligomers. By extending our studies to two additional Aβ receptors, we have revealed common mechanistic principles that have important implications for Aβ neurotoxic signaling, as well as the development of potential therapies for AD.

In our previously published study[22], we showed that PrP, in sub-stoichiometric amounts, potently inhibits the process of Aβ polymerization, as monitored by ThT binding and biochemical assays. Based on mathematical modeling of polymerization kinetics, we demonstrated that PrP$^C$ specifically inhibits the elongation step of Aβ fibril growth, and we suggested that this might result from PrP binding to the growing ends of fibrils, thereby blocking their further elongation. However, this study did not directly measure the effect of PrP on fibril lengths or elongation rates, and did not visualize the localization of PrP on individual fibrils. In the present study, we have greatly extended this previous work using single molecule, super-resolution imaging.

Taken together, our results suggest a model (Fig. 9a) in which PrP$^C$ inhibits the elongation step of Aβ fibril growth by binding specifically to only one end (the more rapidly growing end) of

each fibril, thereby preventing further addition of monomers at that end. Under these conditions, fibril growth can proceed only at the slowly growing end. Multiple lines of evidence reported here support this model. First, fibrils formed in the presence of sub-stoichiometric amounts of PrP are shorter and grow more slowly than under control conditions. Second, PrP causes a significant increase in the total number of fibrils present at any given time. This latter effect is seen with other proteins that inhibit the elongation step of fibril growth, and is predicted by the kinetic models of polymerization based on an increased flux of monomers into secondary nucleation events[35,38]. Third, seeded polymerization experiments, which allow measurement of elongation at the two ends of the fibril separately, demonstrate that PrP completely blocks elongation at one end (the normally fast-growing end), leaving elongation to proceed exclusively at the slow-growing end. Fourth, PrP is localized exclusively at one end of individual Aβ fibrils, and this end corresponds to the fast-growing end under control conditions. Although this last observation makes it unlikely that PrP simultaneously reduces elongation at both fibril ends, real-time growth experiments on single fibrils will be required to definitely resolve this issue.

The model proposed here is consistent with recently published atomic structures of Aβ(1–42) fibrils determined by cryo-EM, solid-state NMR, and X-ray crystallography[56,57]. These studies show that both synthetic and brain-derived fibrils have two structurally distinct ends, based on the binding interface presented to newly added monomers at each end. It was suggested[56] that this structural dimorphism accounts for the polarity of Aβ fibril growth, which was recently demonstrated experimentally[44]. Another, earlier study also presented an Aβ(1–42) fibril structure that displayed a structural polarity[58]. Based on these studies, we postulate that PrP is able to bind specifically to the fast-growing end of the fibril because it recognizes a unique structural interface presented at that end. Our previous study suggested that this is a high affinity interaction with a $K_d$ of 47.6 nM[22]. In that study, we showed that, although the flexible, N-terminal domain of PrP contains both of the identified Aβ binding sites, the structured C-terminal domain is required in order for PrP to inhibit the Aβ polymerization reaction. Thus, both domains of PrP may contribute structurally to interaction with the end of the Aβ fibril. This suggestion is consistent with a recent solid-state NMR study, which demonstrated structural changes in both domains of PrP upon Aβ oligomer binding[59].

There is a great deal of evidence that small oligomeric and protofibrillar forms of Aβ, rather than long amyloid fibrils, are the neurotoxic species primarily responsible for the synaptic loss and cognitive decline in AD[5–7]. We therefore sought to analyze the interaction of PrP with two forms of Aβ, ADDLs and protofibrils, both of which we have confirmed are highly neurotoxic in a dendritic spine retraction assay. Using super-resolution imaging, we localized PrP to one end of short protofibrils, similar to its localization on longer, mature fibrils. On ADDLs, which are globular or ellipsoid in shape without clearly identifiable ends, it was more difficult to resolve the precise location of PrP binding, particularly since these assemblies are at the resolution limit of dSTORM imaging. Nevertheless, we found a statistically significant tendency for PrP to localize eccentrically, toward one edge of the ADDL aggregate. A similar eccentric localization of PrP was observed on small oligomers of Aβ that were normally present during the course of the Aβ polymerization reaction, particularly in the presence of PrP. We postulate that the eccentric localization of PrP on ADDLs and small oligomers may reflect an intrinsic asymmetry in their structure. This suggestion is consistent with the observation that two structurally distinct ends are formed once the fibril reaches a minimum size of six subunits[56]. These considerations raise the interesting possibility

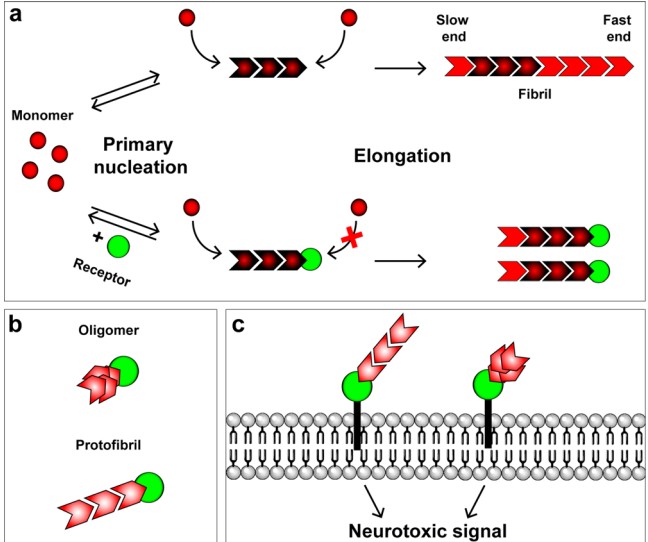

**Fig. 9 Models for the interaction of receptors with Aβ fibrils, protofibrils, and oligomers. a** Schematic showing primary nucleation and elongation steps in the Aβ polymerization process in the absence (upper pathway) and the presence (lower pathway) of a receptor protein, such as PrP$^C$, FcγRIIb, or LilrB2. The receptor binds to the fast-growing end of the fibril, blocking elongation at that end, and restricting elongation to the slow-growing end. Secondary nucleation events are not depicted. **b** Interaction of a receptor with neurotoxic oligomers and protofibrils. Receptors bind to one end/edge of these assemblies, possibly recognizing the same structural determinant present at the fast-growing end of fibrils. **c** Binding of Aβ oligomers and protofibrils to membrane-anchored receptors initiates neurotoxic signaling. Receptors may also trap or concentrate oligomers and protofibrils on the cell surface.

that the structural determinants recognized by PrP on neurotoxic oligomers and protofibrils may be similar to the ones PrP recognizes on the fast-growing end of polymerizing fibrils (Fig. 9b). In this case, the potent neurotoxicity of oligomers would result from the fact that oligomers and protofibrils, in contrast to fibrils, present a high molar concentration of "end-specific" structural determinants to PrP or other cell-surface receptors (Fig. 9c). High-resolution structural studies will be required to define the Aβ-PrP binding interface, as well as the stoichiometry of binding. There is evidence that Aβ protofibrils have a different amyloid core structure than mature fibrils, and may be off-pathway intermediates in the formation of fibrils[60–62]; the same may be true for Aβ oligomers[63,64]. In this case, our results imply that PrP may be capable of trapping neurotoxic forms of Aβ (oligomers and protofibrils) that are normally present in only small amounts during the polymerization process. The SRM data reported here suggest that two or more PrP molecules bind to each Aβ fibril end (Supplementary Fig. 4). This estimate is consistent with biochemical studies, which suggest that the average Aβ oligomer can bind up to six PrP molecules[65].

Most previous literature on the interaction of PrP$^C$ and Aβ has focused on oligomeric forms of Aβ, particularly synthetic ADDLs, since these are the forms thought to be most relevant to neurotoxicity in vivo[10,19–22]. Only a few studies have analyzed the interaction of PrP with Aβ fibrils, or its effect on polymerization or assembly processes. Several different effects of PrP on Aβ aggregation have been described, including inhibition of fibril formation[66], bundling[67] or fragmentation[68,69] of fibrils, and trapping of Aβ in an oligomeric state within PrP-containing complexes[65,68,69]. In a particularly relevant example, it was reported that protofibrils (like those used here) were the most neurotoxic forms of Aβ in an LTP suppression assay, and were the forms that bound most avidly to PrP[21]. However, none of these previous studies localized PrP to particular sites on Aβ assemblies, or identified specific steps in the assembly process that were affected.

Previous studies have identified several other proteins that act as chaperones affecting particular steps in the polymerization pathway of Aβ or other amyloidogenic proteins[35]. For example, Ssa1, an Hsp70-type chaperone in yeast, has been shown to block elongation of fibrils formed by the prion-like protein, Ure2p[70]. Like other biological polymers[71], amyloid fibrils have been shown to elongate in a polarized fashion, with the two ends growing at different rates, reflecting distinct structural interfaces at each end[44–46,72]. PrP is, to our knowledge, the first endogenous factor to be identified that acts primarily as an end-specific inhibitor of amyloid fibril elongation. Interestingly, chaperones that reduce the concentration of oligomeric intermediates generated during polymerization are able to reduce the neurotoxicity of Aβ in biological assays[38,73].

The experiments presented here suggest that two other putative Aβ receptors, FcγRIIb and LilrB2, interact with Aβ via a mechanism similar to that of PrP, involving selective inhibition of elongation. These receptors, which are expressed in both immune cells and neurons, have been shown to bind Aβ oligomers, and their genetic ablation reduces oligomer-induced synaptic toxicity in hippocampal slices[11,12,53]. We found that the extracellular domains of both FcγRIIb and LilrB2, which comprise the known Aβ binding sites, inhibit Aβ polymerization similarly to PrP: in sub-stoichiometric amounts, they increased the polymerization half-time as measured by ThT fluorescence, and they created shorter and more numerous fibrils as monitored by super-resolution imaging. The Aβ binding region of PrP$^C$, which is comprised of two short amino acid motifs within the intrinsically disordered N-terminal domain[10], is structurally unrelated to the Aβ binding regions of FcγRIIb and LilrB2, which are composed of one or two immunoglobulin domains, respectively[11,12]. It remains to be determined how structurally diverse receptors and chaperones selectively recognize localized binding sites on Aβ fibrils and oligomers[35]. One possibility is that Aβ binding induces conformational changes in the receptors that enhance their affinity for specific structural features on these assemblies. Consistent with this idea, NMR studies show that Aβ oligomer binding induces substantial conformational changes in several regions of PrP$^C$ [59,74].

The available evidence suggests that the signal transduction pathways stimulated by Aβ oligomer binding to PrP$^C$, FcγRIIb, and LilrB2 may all be different[12,25,26,53]. Thus, binding of multiple receptors to a common set of structural determinants on Aβ oligomers may activate an array of different signaling mechanisms, which mediate distinct aspects of the synaptotoxic response. PrP$^C$ also serves as a cell-surface receptor mediating the synaptotoxic activity of PrP$^{Sc}$, the infectious form of PrP[51,75], and it has been implicated as a receptor for toxic oligomers composed of tau and α-synuclein[76–79]. It will be of interest to determine whether PrP$^C$ interacts with these other protein aggregates by mechanisms similar to those we have shown here for Aβ, and whether the downstream neurotoxic signaling mechanisms are similar.

Are the interactions between soluble, recombinant PrP$^C$ and Aβ we have documented here relevant in a biological setting, in which most PrP$^C$ is localized on the neuronal cell surface via its GPI anchor? Several pieces of evidence argue that this is, indeed, the case. The fact that exogenous, recombinant PrP blocks the synaptotoxic effect of Aβ oligomers and protofibrils in our hippocampal neuronal assay (Fig. 6, Supplementary Figs. 7 and 8) argues that recombinant and cell-surface PrP$^C$ compete for the same sites on these Aβ assemblies, and that binding to these sites is essential for initiation of a neurotoxic signal (Fig. 9c). Our results raise the possibility that, in addition to serving as a signal-transducing receptor, PrP$^C$ influences the process of Aβ polymerization by stimulating the generation of smaller, more neurotoxic assemblies, an effect we have demonstrated in vitro (Fig. 6). Given the cellular localization of PrP$^C$, it is likely that any effects it has on Aβ polymerization would occur on the plasma membrane of neurons or glial cells in the brain. In this case, PrP$^C$ might serve to trap nascent oligomers or protofibrils on the cell surface, allowing them to accumulate there and initiate PrP$^C$-mediated neurotoxic signaling. Supporting this idea, single-molecule SRM imaging reveals selective association of small Aβ oligomers (dimers and trimers) with PrP$^C$ on the surface of hippocampal neurons[23]. There is also ample biochemical evidence that PrP$^C$ forms complexes with neurotoxic Aβ species in brain tissue from AD patients and transgenic mice, but not in normal brain tissue[74,80–82]. These observations argue that the PrP$^C$-Aβ interactions characterized here are also disease-relevant. Experiments using super-resolution or single-particle microscopy of live cells will be required to address definitively whether glycosylated, GPI-anchored PrP$^C$ on the cell surface interacts differently with Aβ than recombinant PrP. Of note, the fact that PrP promotes formation of smaller, more neurotoxic aggregates of Aβ would argue against the proposed strategy of using soluble, recombinant PrP as a drug to treat AD[83].

Current therapies for AD are focused primarily on lowering levels of Aβ, either by inhibiting its synthesis or enhancing its degradation[84]. These therapies have met with little success in recent clinical trials. The results presented here raise the possibility of a therapeutic approach based on blocking interactions of neurotoxic forms of Aβ with its cellular receptors, thereby reducing the trapping of toxic oligomers and protofibrils on the neuronal surface, and inhibiting activation of downstream signaling pathways engaged by these receptors. Indeed, there is evidence that PrP$^C$- directed ligands (small molecules and

antibodies) can have positive therapeutic effects in AD mouse models[31,32,85]. We suggest that it will be possible to fine-tune this approach, based our observation that there is a common structural interface on Aβ fibril ends, as well as on neurotoxic protofibrils and oligomers, that is recognized by multiple Aβ receptors. By defining this interface at the atomic level, it may be feasible to design small molecules that block interaction of neurotoxic Aβ species with all of these receptors simultaneously, thereby providing a highly efficient AD therapeutic. This structural information may also inform creation of diagnostic reagents specific for neurotoxic forms of Aβ.

## Methods

**Preparation of Aβ monomers**. Lyophilized human Aβ (1–42), Aβ-Cy5 (1–42), and Aβ-Cy3 (1–42), were synthesized by ERI Amyloid Laboratory, LLC (Oxford, CT, USA). Maleimide derivatives of Cy3 and Cy5 were conjugated to an acetylated cysteine residue included at the N-terminus of Aβ (1–42). Details of monomer preparation can be found in Bove-Fenderson, et al.[22]. Unlabeled monomers were solubilized in 15 mM NaOH, and were then isolated by size exclusion chromatography on a Superdex 75 10/300 GL (GE Healthcare) column using PBS as the running buffer. Fractions were collected and were immediately used in ThT assays. Cy3- and Cy5-labeled peptide was solubilized in 15 mM NaOH and was used directly for ThT assays and super-resolution microscopy. The concentration of Aβ was estimated with a NanoDrop UV-visible spectrometer (Thermo Scientific) by reading the sample absorbance at 214 nm and applying Beer's Law with an extinction coefficient of 76,848 $M^{-1}$ $cm^{-1}$.

**Preparation of ADDLs and protofibrils**. Fluorescently labeled ADDLs were prepared using a standard protocol[86,87] in which lyophilized Aβ peptide was solubilized in HFIP and then dried to a film. The ratio between labeled and unlabeled peptide was 1:10. The film was then solubilized in DMSO before dilution to a concentration of 100 μM in phenol red-free Ham's F12 medium (DMSO 2% v/v), followed by incubation at room temperature for 16 h. To prepare protofibrils[21], ADDL samples were incubated at room temperature for either 3 days (protofibril preparation 1) or 7 days (protofibril preparation 2).

**Recombinant PrP**. Full-length mouse PrP (23-230) was produced and purified as described previously[22]. E. coli strain BL21 Star was transformed with the pJ411 vector expressing murine PrP23-230. Cells were lysed, and then PrP was purified with an ÄKTA purification system (GE Healthcare) using a $Ni^{2+}$-immobilized metal ion affinity column. Protein was eluted from the $Ni^{2+}$ immobilized metal ion affinity column with 5 M guanidine HCl, 0.1 M Tris acetate, 0.1 M potassium phosphate (pH 4.5) while monitoring $A_{280}$. Fractions spanning the elution peak were combined, and the pH was raised to 8 by titration with potassium acetate. The pooled samples were then desalted into 20 mM potassium acetate, pH 5.5 using a HiPrep 26/10 desalting column (GE Healthcare), and PrP was purified by reverse-phase HPLC using a C4 column (Grace/Vydac). Fractions containing the purified protein were pooled, lyophilized, and stored at −80 °C for future use.

For fluorescent labeling, PrP was prepared with a cysteine residue substituted for a glycine residue at position 34 (G34C). Lyophilized PrP G34C was dissolved in 20 mM potassium acetate, pH 5.5 to a concentration of 100 μM. Alexa Fluor 555 or 488 C2 maleimide (ThermoFisher Scientific) was added dropwise with stirring from a stock solution of 1 mM in water, to a final ratio of 1:10 (protein:dye). This solution was incubated at room temperature for two hours. One ml of the solution was then injected into an analytical C3 column (Zorbax 300SB C3, Agilent) on an Agilent 1200 Infinity HPLC system, and the peptide peak/dye was collected and lyophilized. Confirmation of successful linkage was made by MALDI-TOF mass spectrometry.

**Other recombinant proteins**. Recombinant FcγRIIb and LilrB2 (extracellular domains) were purchased from Novoprotein (C444) and R&D systems (8429-T4), respectively.

**ThT assay for Aβ polymerization**. Kinetic assays for Aβ polymerization were conducted as described previously[22,34,43]. Aβ monomers were diluted to a concentration of 5–20 μM in PBS, and 10 μM ThT was added. Recombinant proteins were added from a 1 mg/ml stock in water at the indicated concentrations. To follow ThT binding, 100 μl samples were placed in 96-well, half-volume, low-binding plates (Corning 3881), and fluorescence was read in a Synergy H1 Multi-Mode Microplate Reader (BioTek) every 2 min at 37 °C (excitation 440 nm, emission 480 nm).

**Preparation of Aβ samples for super-resolution microscopy**. Fluorescently labeled Aβ fibrils were formed by polymerizing 100% Cy5-labeled Aβ 1–42 monomers. Labeled monomers were diluted to a concentration of 5–20 μM in PBS, followed by incubation at 37 °C for 24 h to 1 week. Where indicated, recombinant

proteins were added to monomeric solution at the starting point of the polymerization reaction.

For seeding assays, fibrils prepared from Aβ-Cy5, as described above, were diluted in PBS to a monomer-equivalent concentration of 10 μM, and were sheared by sonication (30 s on a 50% duty cycle, Branson 1800) to yield seeds. To initiate seeded growth, freshly prepared Aβ-Cy3 monomer (10 μM) was added to an equal volume of the seed solution, and incubated for 24 h at 37 °C.

For three-color imaging, fibrils were first immobilized on antibody-coated wells. Glass-bottom, multi-well plates (Lab-Tek) were sequentially cleaned in 1 M HCl, 70% ethanol, and 1 M KOH. After extensive rinsing with ultrapure water, plates were dried with $N_2$. The plates were then treated with an antibody against amyloid-β (6E10, mouse monoclonal primary, BioLegend, Cat. #: 803001) overnight. Seeded fibrils, prepared as described above, were incubated with PrP-AF488 for 30 min at 37 °C, and were then added to the antibody-coated wells to allow the fibrils to adhere. After 30 min of incubation, wells were washed with ultrapure water to remove unbound PrP.

**Super-resolution imaging (SIM and dSTORM)**. Super-resolution microscopy was performed at the Harvard Center for Biological Imaging (HCBI) (Cambridge, MA, USA) using a Zeiss ELYRA microscope, which is capable of performing both dSTORM and SIM imaging. This microscope is equipped with 488 nm, 561 nm, and 638 nm laser lines, and a ×100 oil immersion objective lens (NA 1.4). For SIM imaging Aβ samples were dried onto pre-washed coverslips and covered in mounting medium (Vectashield H1000, Vector laboratories). For dSTORM imaging, samples were dried onto glass-bottom multi-well plates (Lab-Tek), and were then covered in photoswitching buffer solution immediately before imaging; this solution consisted of 100 mM mercaptoethylamine (MEA) in phosphate-buffered saline (PBS, pH 7.4), together with a glucose-enzyme oxygen scavenger (40 mg/ml glucose, 50 mg/ml glucose oxidase, 1 mg/ml catalase). The chamber was filled to the top, and was closed with a cap to minimize entrance of oxygen. Collected images were then processed using ZEN 2.3 software.

**Image processing**. We developed a MATLAB code to quantitate the size and number of Aβ aggregates (fibrils, protofibrils, and ADDLs) in super-resolution images. In the first step, grayscale SIM images imported to MATLAB and were smoothed by using 2 × 2 unit square kernel. The smoothing process helps to reduce noise within an image. The smoothed images were then binarized and converted into black-and-white image by using a threshold greater than 2× the standard deviation of the pixel value distributions. The binarization process generates sharp boundaries for each object, which were then detected by using the 'boundary' function of MATLAB. The length of each aggregate was defined by determining the maximum distance between pairs of points on the detected boundary, with these points being used to define the two ends of the aggregate. We calculated the cumulative distribution of aggregate lengths, the mean length for the distribution, and the mean number of aggregates/μm² in each SIM image, and took the mean values from multiple images to arrive at the final values reported in the figures.

In the case of two-color SIM images, PrP dimensions and positions were determined as described above for Aβ aggregates. The colocalization between PrP and Aβ was then computed by counting the number of PrP pixels that overlapped with Aβ aggregate pixels. This number was then normalized to total number of PrP pixels in the SIM image.

In order to quantify the localization of PrP with respect to Aβ fibril ends, we measured the minimum distance ($D_{min}$) between each fluorescent PrP spot and the closest end of the associated fibril. This distance was then normalized to the total length of the fibril ($D_{min}/L_{total}$). Random distributions were generated by using unrelated image of Aβ and PrP from two different experiments.

**Synaptotoxicity assay**. All procedures involving animals were conducted according to the United States Department of Agriculture Animal Welfare Act and the National Institutes of Health Policy on Humane Care and Use of Laboratory Animals. Hippocampal neurons were cultured from P0 pups (C57BL6 mice, both sexes) as described[51,88]. Neurons were seeded on poly-L-lysine-coated coverslips, and after 24 h the coverslips were inverted onto an astrocyte feeder layer and maintained in NB/B27 medium until used. The astrocyte feeder layer was generated using P0 cerebral cortex. Neurons were kept in culture for 21 days prior to Aβ treatment.

Neurons were treated for 1 h with vehicle, or with 500 nM (monomer equivalent) of ADDLs, protofibrils, fibrils, followed by fixation in 4% paraformaldehyde and staining with Alexa 488-phalloidin (ThermoFischer Scientific, Waltham, MA, Cat. #: A12379) to visualize dendritic spines. In some experiments (Fig. 6), Aβ preparations were incubated with recombinant PrP before addition to neuronal cultures. Images were acquired using a Zeiss 880 confocal microscope with a 63x objective (N.A. = 1.4). The number dendritic spines per μm of dendrite length was determined using ImageJ software, as described previously[51].

**Electron microscopy**. Samples of ADDLs, protofibrils, and Aβ fibrils were prepared as described above. The samples were then applied as a 5 μl droplet to a glow-discharged, 300-mesh copper grids and allowed to incubate for 3 min before

washing 3 times with filtered, ultrapure water. The grid surface was then stained for 1 min in 2% uranyl acetate and dried for 5 min. Images were taken using a Philips CM12 120KV transmission microscope. Size measurements of aggregates were made using ImageJ.

**Statistical analysis**. Data are shown as mean ± S.E. Statistical significance of the differences between mean values was evaluated using the two-sided Student's *t*-test. For cases in which there were multiple comparisons, we used the Holm–Bonferroni correction to adjust the *P*-values. An adjusted *P*-value of <0.05 was considered to be statistically significant. The number of biological independent samples in each experiment is ≥3 unless otherwise is specified in the figure legend.

**Reporting summary**. Further information on research design is available in the Nature Research Reporting Summary linked to this article.

## Data availability

The datasets generated and analyzed during the current study are available from the corresponding author upon a reasonable request. Source data are provided with this paper.

## Code availability

The analysis codes used in this study are available from the corresponding author upon request.

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

## Acknowledgements

This work was supported by NIH grant R01 NS065244 to D.A.H.; and an Alzheimer's Association Research Fellowship (2018-AARF-589708) to L.A.. SRM was performed at the Harvard Center for Biological Imaging (HCBI) (funded by NIH grant 1S10RR029237-01). We thank Douglas Richardson at the HCBI for his advice on SRM.

## Author contributions

D.A.H. and L.A. conceived the project. L.A. performed the experiments. L.A. and D.A.H. designed the experiments, analyzed the data, and wrote the paper.

## Competing interests

The authors declare no competing interests.
