## [Peer Review File · Nature Communications]

Reviewers' comments:

Reviewer #1 (Remarks to the Author):

The paper by Amin L. & Harris D. submitted for publication at Nat. Commun. is a follow up of a previous study by the same group indicating that recombinant cellular prion protein PrP targets amyloid- β oligomers and fibril ends and prevents their elongation (Bove-Fenderson, 2017, JBC). Based on in vitro experiments, Amin's paper documents that the inhibitory effect of PrP on A β fibril polymerization is due to specific PrP binding at the fast-growing end of A β fibrils. Compared to their previous work, well-designed super-resolution microscopy experiments allowed measuring the inhibition rate of A β fibril polymerization by PrP and the mean length of generated fibrils. The authors further show the regionally binding of PrP on other A β assemblies such as ADDL or protofibrils. Still using recombinant proteins, the authors show in vitro that other putative receptors for A β oligomers/fibrils, i.e., FcyRIIb and LILRB2, also exert an inhibitory effect on A β fibril polymerization (with quite less efficacy than PrP). Inhibiting the polymerization of A β increases the proportion of small A β oligomers that are described as highly neurotoxic entities compared to larger fibrils or aggregates. While potentially interesting in the context of Alzheimer's therapeutics, the manuscript should be deeply improved to meet the standards of Nature Communications.

1) The manuscript suffers somewhat from overinterpretation of the data. For example, in the third paragraph of the result section "A β polymerization is strongly polarized, and PrP selectively blocks elongation at the more rapidly growing end", the authors concluded that PrP completely blocks elongation at the fast-growing end of the fibril. This conclusion is however not fully supported by the experiments. Measuring in the presence of PrP an elongation rate of the A β fibril at one extremity similar to that of the slow-growing end (end 2) in the absence of PrP does not permit to conclude that PrP only affects the fast-growing end (Figure 3). With the data presented in Figure 3, how can the author formally exclude that PrP does not impact on both A β fibril extremities with the fast-growing rate of end 1 (red line) decreasing coincidentally to a slow-growing rate (black dashed line) and the slow-growing rate of end 2 (red dashed line) becoming quite negligible (black line)? The authors need to be more accurate in how they state their conclusions so that they match the data presented.

2) PrP binds at the fast-growing end of the A β fibril. The author should estimate the number of PrP molecules bound at this extremity necessary to inhibit the A β polymerization process. This would also be important because of PrPC-mediated toxicity of A β oligomers/fibrils in neurons.

3) The inhibition of A β fibril elongation by PrP (or FcyRIIb and LILRB2) promotes a rise of small A β oligomers. The fact that these generated, small A β oligomers are potentially neurotoxic is however not established in this study. The authors have to assess the neurotoxicity of these A β species after in vitro receptor-mediated inhibition of A β fibril elongation using, for example, their cell-based assay with primary hippocampal neurons. Supplementary figure 5 only displays results with ADDL, protofibril preparations, and fibrils that were not pretreated with recombinant PrP or other receptors.

4) A major criticism of this paper is that the inhibitory effect of PrP (and other putative receptors) on A β polymerization is observed in vitro with recombinant full-length PrP produced in *E. coli*. Do the authors have any ex vivo/in vivo evidence that cell surface PrPC does exert the same inhibitory effect on A β polymerization notably in the context of Alzheimer's disease?

5) Connected to point 4, would all PrPC isoforms (non, mono, biglycosylated PrPC) display the same capacity to bind A β oligomers and block their elongation?

6) As several putative A β receptors display an inhibitory effect on the elongation of fibrils of A β , what could be the structural determinant(s) common to those proteins that recognize(s) the fast-growing end of the A β fibrils?

7) How would the authors reconcile their data about the inhibitory effect of PrPC on A β polymerization with the presence of senile plaques in the Alzheimer's brain as A β plaques originate from the previous polymerization of A β peptides and subsequent deposition?

Reviewer #2 (Remarks to the Author):

The authors report super-resolution analyses of interactions between A-beta assemblies, PrPC and two other receptors for A-beta assemblies. They demonstrate clearly the polarized interaction of PrPC with the more rapidly growing ends of A-beta oligomers, protofibrils and fibrils. This interaction blocks the further growth of these assemblies. The findings are novel and relevant to understanding the how pathological A-beta species interact with receptors that mediate neurotoxicity in the context of Alzheimer's disease. The presentation is clear and illuminating. The data are visually stunning, mechanistically revealing, and supportive of the testing therapeutic strategies that target these interactions (with the caveat noted below). I found little to criticize. However, regarding therapeutic strategies, I am not sure that it is easy to predict the in vivo consequences of blocking the binding of A-beta species with their receptors. On the face of it, as the authors argue, one might expect an immediate reduction in neurotoxic signal transduction induced by existing A-beta assemblies. On the other hand, if receptor interactions block the growth of pathological A-beta species as the authors have shown, then blocking the interactions might promote the accumulation of A-beta multimers. Granted, the longer the fibrils the less toxic they appear to be per particle, but fibrils still have some toxicity, and if there are proteostatic mechanisms that fragment them, a treatment that blocks the ability of receptors such as PrPC to reduce A-beta fibrilization might have a detrimental effect. Of course, we won't know whether the potentially beneficial or detrimental effects of such a treatment would win the day until appropriate trials are done.

Reviewer #3 (Remarks to the Author):

In this manuscript, Amin and Harris, provide novel and important details about the mechanism and molecular consequences of PrPc binding to A β species. In the literature A β has been described to bind many cellular receptors with PrPc A β interaction observed by multiple labs. Blocking the binding of Abeta to PrPc has been shown to be beneficial in mouse models of AD. But until now, the precise mechanisms of PrPc binding and the molecular consequences were not clear.

The authors demonstrate that PrPc inhibits A β fibril growth in an in vitro fibrilization assay in a dose-dependent manner. They also demonstrate that A β fibrils grow in a polarized manner and PrPc specifically blocks growth at the fast-growing end. They demonstrate highly preferential binding to the fast-growing end, where it appears to block any further addition of

subunits. They also looked at PrPc localization with smaller A β species, ADDLs and protofibrils, and found a similar, asymmetric localization, suggesting these species exhibit polarity as well. Two other well studied receptors Fc γ RIIb and LILrB2, also showed the same effect on polymerization kinetics and fibril size as PrPc.

The super resolution microscopy (SIM and dSTORM) provide very high resolution (100nm and 20nm respectively) images that visualize well the properties described, and the quantification is thorough and convincing. The analysis is very informative and the finding is novel. It is useful to know this structural information for designing molecules to inhibit the binding. The authors' techniques could be used to develop a good assay for screening small molecule inhibitors of PrPc and other receptor binding to A β fibrils, the ultimate development of which could provide therapeutic benefit without complete understanding of how PrPc binding to A β causes toxicity.

Comments:

1. The main question that comes to mind centers on the physiological relevance of this biochemical process observed in vitro. Does this effect of PrPc and other receptors on A β fibril formation in isolated highly simplified in vitro systems actually affect disease processes in the human brain? The μ M concentrations of PrPc and A β used may not be physiological in most areas of the brain, and PrPc and the other receptors are mainly cell surface bound rather than diffusing freely.
2. What is causing the toxic effect of the PrPc interaction with amyloid? Is it the fact that PrPc (or LILrB, etc.) prevents the growth of the long end of amyloid fibrils creating greater numbers of smaller, more toxic species that cause greater cellular damage? This could happen if soluble species of PrPc are circulating and disrupting fibrilization. Or is it simply that amyloid fibril binding the PrPc, LILrB2 or Fc γ RIIb activates various toxic intracellular signaling cascades, and the effect on amyloid fibrilization is just a consequence of preferential binding, but not really relevant to disease processes? Or may both pathways be disease relevant?
3. Related to #1 above, cell culture or in vivo experiments should be performed to illustrate the physiological consequences of this biochemical mechanism. Perhaps a mutant PrPc that still binds amyloid fibrils and affects fibril growth, but has lost intracellular signaling capabilities could differentiate effects. Or amyloid oligomers could be added to primary cultures in the presence and absence of recombinant PrPc followed by a toxicity assay, such as measurement of dendritic spine density. If blocking fibrilization increases toxicity, then soluble PrPc would increase toxicity. Alternatively, if binding to an Abeta receptor caused toxicity, then soluble PrPc should be protective. It would also be informative to Abeta-treat PrPC^{-/-} neurons, which should be resistant to Abeta toxicity.

Reviewer #4 (Remarks to the Author):

The manuscript by Amin and Harris analyzes the effect of the prion protein PrP on the fibril elongation of the amyloid-beta peptide (Abeta), which forms amyloid plaque in Alzheimer's disease. Oligomeric forms of Abeta have been implicated as the main neurotoxic agent in AD, but the process of neurotoxicity is poorly understood. Membrane receptors have been implicated in the uptake of Abeta, and membrane bound PrP is hypothesized as a potential modulator of Abeta toxicity. Here, the authors use super-resolution microscopy to visualize the interaction of PrP with Abeta oligomers and fibrils. They find that PrP specifically binds to

fibril ends, inhibiting their growth and shifting Abeta aggregate populations towards smaller, more toxic species. The authors hypothesize that two membrane receptors FcγRIIb and LILRB2, which also inhibit fibril elongation, act by the same mechanism as PrP. The study offers an intriguing model how cellular factors can affect Abeta aggregation and how, counterintuitively, inhibitors of fibril elongation may exacerbate amyloid toxicity. This interaction may present a new type of therapeutic target for a class of disease for which effective therapies have proven to be elusive.

The authors find that PrP molecules cap Abeta fibrils at the fast growing fibril end and thus shift the size distribution of towards oligomers and short fibrils. This coincides with an increased Abeta toxicity when these fibrils are added to neuronal culture. This is an highly suggestive result, however, the study does not quite connect the in vitro data with the relevance in vivo. The authors use fluorescently labeled Abeta and recombinant PrP from *E. coli* to analyze the interaction. However, recombinant PrP lacks the GPI anchor that tethers the protein to the membrane as well as the glycosylation of the mammalian prion protein. This raises several questions the study would need to answer:

1) What is the aggregation state of the PrP added to the assay? Is it the monomeric protein or a preformed PrP aggregate? The authors need to show data characterizing the aggregation / oligomerization of PrP by itself under the assay conditions. The underlying hypothesis is that membrane-bound PrP monomers interact with the nascent fibril. However, whether this is the case in vitro, is not shown.

2) Does membrane bound PrP on the cell membrane cap the growth of Abeta fibrils the same way the soluble recombinant protein does? Expression of GPI-less PrP in vivo produces a very different prion disease phenotype from normal membrane-bound PrP, so it is likely that the molecular properties of both forms are quite different.

3) What are the binding constants of PrP to Abeta oligomers and fibrils, respectively? This is an important piece of data to judge whether the interaction observed in vitro is plausible under physiological conditions.

The authors then look at the evolution of Abeta oligomers (ADDL) into protofibrils and fibrils and find that PrP shifts the population towards shorter, more toxic species, most likely by the fibril end-capping mechanism characterized in Fig 3. This is a very nice result, but it raises a number of technical questions, especially about the discrimination between oligomers, protofibrils and fibrils and whether the aggregates size distribution under conditions of the toxicity assay is the same as it was in vitro:

4) How is a protofibril discriminated from an ADDL, and how is a protofibril distinguished from a fibril? It seems that the length distribution of protofibrils extends into objects of several micrometers, which is a typical length of fibrillar Abeta. Superresolution images of panels labeled as protofibrils show straight fibril-shaped objects. Typically, protofibrils are defined as worm-like, short linear assemblies that are distinct in diameter and shape from straight amyloid fibrils. The data shown here give no indication that the object imaged are indeed protofibrils. This claim needs to be backed up by AFM or EM imaging and a clear criterion for distinguishing both types of assemblies needs to be established.

5) S4 the length distribution of protofibrils in the panel B of Fig S4 look very heterogeneous, which is also reflected in the distribution of panel E. However, panel D shows a narrow distribution with tight error bars. Please explain this apparent discrepancy.

6) Similarly, panels F-I in S5 show a distribution of spherical objects (presumably ADDL) and fibril-shaped objects of varying length. It is not clear to me how protofibrils fit into this picture. The data could easily be explained by a conversion of oligomers (ADDL) into fibrils. It has been well established that oligomers are the main toxic Abeta species, so that their depletion would explain the observed reduction in toxicity that matches the disappearance of oligomeric species and the growth of fibrils.

7) The length distribution of fibrils and 3 d protofibrils used in toxicity assays is missing.

8) The method section states that for toxicity assays, neurons were treated with 500 mM Abeta. This is clearly wrong as the Abeta peptide is not soluble at 0.5M concentration. Did the authors mean 500 μ M? Even that would be a huge concentration, which would probably lead to uncontrolled aggregation of the peptide in the cell culture medium. The authors need to show that the size distribution of Abeta aggregates encountered by the neurons is actually the same that they characterized in their in vitro aggregation assays.

9) Minor point: Scale bars are missing in S4panels B-C, scale bar is not defined in length in panel A. Scale bars missing in all panels of Fig S5 neuronal images.

Finally, the authors analyze the effect of the two membrane receptors Fc γ RIIb and LILRB2 on Abeta fibril growth. Analysis of bulk aggregation finds that both proteins (or more specifically their extracellular domains) inhibit aggregation in vitro. Both protein also shift fibril size distribution towards shorter species, similar to PrP. This is a very suggestive result. However, it raises the same conceptual and technical issues just discussed for PrP:

10) The authors need to show that the proteins actually interact with the growing fibril ends of Abeta.

11) What is the aggregation state of the proteins?

12) How can we be sure that the membrane-associated native form of the proteins has the same effect on Abeta aggregation as the extracellular domain fragment?

MS ID#: NCOMMS-19-34520B

MS TITLE: A β receptors specifically recognize molecular features displayed by fibril ends and neurotoxic oligomers

AUTHORS: Ladan Amin and David A. Harris

Reviewer #1 (Remarks to the Author):

1) The manuscript suffers somewhat from overinterpretation of the data. For example, in the third paragraph of the result section “A β polymerization is strongly polarized, and PrP selectively blocks elongation at the more rapidly growing end”, the authors concluded that PrP completely blocks elongation at the fast-growing end of the fibril. This conclusion is however not fully supported by the experiments. Measuring in the presence of PrP an elongation rate of the A β fibril at one extremity similar to that of the slow-growing end (end 2) in the absence of PrP does not permit to conclude that PrP only affects the fast-growing end (Figure 3). With the data presented in Figure 3, how can the author formally exclude that PrP does not impact on both A β fibril extremities with the fast-growing rate of end 1 (red line) decreasing coincidentally to a slow-growing rate (black dashed line) and the slow-growing rate of end 2 (red dashed line) becoming quite negligible (black line)? The authors need to be more accurate in how they state their conclusions so that they match the data presented.

The reviewer is entirely correct on this point. Because the data in Fig. 3 represent ensemble measurements of a population of different fibrils, they demonstrate only that PrP alters elongation rates at fibril ends, but do not allow direct estimation of elongation rates at the two ends of individual fibrils. Thus, as the reviewer points out, one cannot rule out the possibility that PrP is simultaneously altering growth rates at both ends of the fibril. However, we wish to point out that the data in Figs. 4 and S3, which show that PrP binds selectively to the fast-growing end of individual fibrils, makes this scenario unlikely, since it would require that binding of PrP to the fast-growing end of the fibril indirectly alters growth at the opposite (slow-growing) end, which can be many microns away, depending on the length of the fibril. The only way to formally rule out this possibility, however, would be to directly measure growth rates at the two ends of individual fibrils over time. We have begun to establish the capability of performing such real-time growth measurements on individual fibrils, but believe that such experiments are properly the subject of a subsequent publication, since they involve detailed kinetic and thermodynamic analyses, which are beyond the scope of the present publication. Nevertheless, to explicitly acknowledge the limitations of the data as shown, we have qualified our conclusions as follows

[From Results, p. 5]:

...However, because the data shown in Fig. 3 represent ensemble measurements of a population of fibrils, we cannot formally rule out the possibility that PrP is simultaneously altering growth

rates at both ends of the fibril. We regard this possibility as unlikely, however, based on the next set of experiments, in which we directly visualized the location of PrP on individual A β fibrils.

PrP binds exclusively to the fast-growing end of A β fibrils

A more likely mechanism by which PrP blocks fibril elongation is by binding selectively to the fast-growing end of the fibril, preventing further monomer addition at that end, without any effect on elongation at the slow-growing end. To directly localize PrP on individual fibrils....

[From Results, p. 7]:

...These data indicate that PrP inhibits fibril elongation by binding selectively to the fast-growing end of the fibril, thereby blocking growth at that end. The localization of PrP exclusively at the fast-growing end of the fibril makes it unlikely that PrP also affects fibril elongation at the slow-growing end, which can lie many microns away, depending on the length of the fibril. Real-time growth experiments on individual A β fibrils using SRM will allow definitive resolution of this question.

[From Discussion, p. 11]:

Although this last observation makes it unlikely that PrP simultaneously reduces elongation at both fibril ends, real-time growth experiments on single fibrils will be required to definitely resolve this issue.

2) PrP binds at the fast-growing end of the A β fibril. The author should estimate the number of PrP molecules bound at this extremity necessary to inhibit the A β polymerization process. This would also be important because of PrPC-mediated toxicity of A β oligomers/fibrils in neurons.

We agree with reviewer that this is an important question, and we have considered several ways of determining the binding stoichiometry. Estimating the number of PrP molecules that bind to a single A β aggregate based on the initial concentrations of the two species is not possible, since A β preparations are extremely heterogeneous in terms of size. Using the fluorescent brightness of PrP accumulations at the ends of individual fibrils might be feasible, but converting brightness to absolute molecular concentrations is subject to a number of imaging-related artifacts (Khater, Nabi et al., *Patterns* 1, no. 3 2020). Therefore, we decided to address the stoichiometry question in an experiment using a 50:50 mixture of PrP molecules labeled with either one of two different fluorophores (PrP-AF555 and PrPAF488). This experiment allowed us to determine that each fibril end can bind two or more PrP molecules. We have added a new supplementary figure to the revised version of the manuscript describing this experiment (new Figure S4), and the following paragraphs were inserted into the Results and Discussion sections:

[From Results, p. 7]:

Finally, we wished to investigate the stoichiometry of PrP binding to fibrils ends. In order to determine whether each fibril end bound one or more molecules of PrP, we performed a triple-label experiment (Supplementary Fig. S4) in which A β -Cy5 monomers were incubated for 24 h in the presence of an equimolar (50:50) mixture of PrP labeled with either Alexa Fluor 488 (PrP-AF488) or Alexa Fluor 555 (PrP-AF555). Samples were then imaged with three-color SIM. In each image, we measured the number of A β -associated PrP clusters containing either AF488 or AF555, as well as the number of PrP clusters containing both fluorophores. We determined that 58 \pm 0.03% of the fibril-associated PrP clusters were labeled with both AF488 and AF555, indicating that the majority of clusters contained more than one molecule of bound PrP. Several such clusters are indicated by arrows in Fig. S4A. These data indicate that most fibril ends bind more than one PrP molecule. We also found that only 25 \pm 0.03% of the PrP clusters not associated with fibrils contained both labels, indicating that most unbound PrP molecules were monomeric. This result argues against artifactual aggregation of PrP during the course of the experiment.

[From Discussion, p. 12]:

High-resolution structural studies will be required to define the A β -PrP binding interface, as well as the stoichiometry of binding. There is evidence that A β protofibrils have a different amyloid core structure than mature fibrils, and may be off-pathway intermediates in the formation of fibrils⁶⁰⁻⁶²; the same may be true for A β oligomers^{63,64}. In this case, our results imply that PrP may be capable of trapping neurotoxic forms of A β (oligomers and protofibrils) that are normally present in only small amounts during the polymerization process. The SRM data reported here suggest that two or more PrP molecules bind to each A β fibril end (Supplementary Fig. S4). This estimate is consistent with biochemical studies, which suggest that the average A β oligomer can bind up to six PrP molecules⁶⁵.

3) The inhibition of A β fibril elongation by PrP (or FcyRIIb and LILRB2) promotes a rise of small A β oligomers. The fact that these generated, small A β oligomers are potentially neurotoxic is however not established in this study. The authors have to assess the neurotoxicity of these A β species after in vitro receptor-mediated inhibition of A β fibril elongation using, for example, their cell-based assay with primary hippocampal neurons. Supplementary figure 5 only displays results with ADDL, protofibril preparations, and fibrils that were not pretreated with recombinant PrP or other receptors.

We have performed the experiment suggested by the reviewer, and have presented the results in two new figures (Figs. 6 and S8). These are included in a new sub-section of the Results entitled “Effect of PrP on neurotoxicity of A β assemblies”.

[From Results, p. 9]:

Effect of PrP on neurotoxicity of A β assemblies

... We then performed a second kind of experiment to investigate how the toxicity of A β is affected by polymerization in the presence of PrP. Since inclusion of PrP in the polymerization reaction results in the accumulation of large numbers of short A β fibrils (Fig. 1), we asked whether these short fibrils were neurotoxic. In these experiment, we first polymerized A β monomers (5 μ M) in the presence of increasing concentration of PrP (0, 0.1, 0.5 and 1 μ M) for 24 h (Fig. 6A). The A β aggregates that formed were then added to cell media at a final concentration of 500 nM (A β monomer-equivalents) and incubated for 24 h. We found that A β samples polymerized with PrP were highly neurotoxic, and their toxicity increased as the concentration of PrP present in the reaction increased (Fig. 6B-D), correlating with the presence of shorter and more numerous fibrils (Fig. 1). In addition, we observed that the toxicity of these short fibrils was blocked by addition of excess recombinant PrP (5 μ M) to the polymerization reaction, prior to dilution into tissue culture medium for treatment of the neurons (Fig 6E-F). This rescuing effect was also observed when fibrils were polymerized using a higher concentration of A β (20 μ M) (Supplementary Fig. S8). This experiment demonstrates that the presence of PrP during the polymerization reaction favors accumulation of short, highly neurotoxic fibrils, but that saturation of these fibrils with additional PrP reduces their toxic effect.

4) A major criticism of this paper is that the inhibitory effect of PrP (and other putative receptors) on A β polymerization is observed *in vitro* with recombinant full-length PrP produced in *E. coli*. Do the authors have any *ex vivo/in vivo* evidence that cell surface PrP^C does exert the same inhibitory effect on A β polymerization notably in the context of Alzheimer's disease?

We have now inserted a new paragraph in the Discussion that specifically addresses this point. We cite literature studies that support a specific interaction between PrP^C and A β aggregates *ex vivo* on the surface of cultured neurons, as well as *in vivo* in brain tissue from AD patients and transgenic mice. We are planning our own SRM experiments to investigate how membrane-anchored PrP^C influences A β polymerization and the formation of A β oligomers on the neuronal surface, but these will be the subject of a subsequent paper.

[From Discussion, p. 13]:

Are the interactions between soluble, recombinant PrP^C and A β we have documented here relevant in a biological setting, in which most PrP^C is localized on the neuronal cell surface via its GPI anchor? Several pieces of evidence argue that this is, indeed, the case. The fact that exogenous, recombinant PrP blocks the synaptotoxic effect of A β oligomers and protofibrils in our hippocampal neuronal assay (Figs. 6, S7, and S8) argues that recombinant and cell-surface PrP^C compete for the same sites on these A β assemblies, and that binding to these sites is essential for initiation of a neurotoxic signal (Fig. 9C). Our results raise the possibility that, in addition to serving as a signal-transducing receptor, PrP^C influences the process of A β polymerization by stimulating the generation of smaller, more neurotoxic assemblies, an effect we have demonstrated *in vitro* (Fig. 6). Given the cellular localization of PrP^C, it is likely that any effects it has on A β

polymerization would occur on the plasma membrane of neurons or glial cells in the brain. In this case, PrP^C might serve to trap nascent oligomers or protofibrils on the cell surface, allowing them to accumulate there and initiate PrP^C-mediated neurotoxic signaling. Supporting this idea, single-molecule SRM imaging reveals selective association of small A β oligomers (dimers and trimers) with PrP^C on the surface of hippocampal neurons²³. There is also ample biochemical evidence that PrP^C forms complexes with neurotoxic A β species in brain tissue from AD patients and transgenic mice, but not in normal brain tissue^{74,80-82}. These observations argue that the PrP^C-A β interactions characterized here are also disease-relevant. Of note, the fact that PrP promotes formation of smaller, more neurotoxic aggregates of A β would argue against the proposed strategy of using soluble, recombinant PrP as a drug to treat AD⁸³.

5) Connected to point 4, would all PrPC isoforms (non, mono, biglycosylated PrPC) display the same capacity to bind A β oligomers and block their elongation?

Glycosylated PrP^C can certainly bind A β , as demonstrated by the studies cited above, which use brain- or cell-expressed PrP^C. Whether different glycoforms show differential binding capacities has never been investigated. The two canonical binding sites for A β oligomers are in the N-terminal domain, while the two N-linked glycosylation sites are in the C-terminal domain. However, as discussed in the paper, we have shown previously that the C-terminal domain is essential for the inhibitory effect of PrP on A β polymerization²², so it is conceivable that glycosylation could affect the interaction of PrP^C and A β .

6) As several putative A β receptors display an inhibitory effect on the elongation of fibrils of A β , what could be the structural determinant(s) common to those proteins that recognize(s) the fast-growing end of the A β fibrils?

A definitive answer to this question must await high-resolution structural studies of A β fibrils bound to PrP and other receptors. We are currently pursuing such studies using cryo-EM and other techniques. However, we have added text to the Discussion section addressing this question, and speculating on a possible mechanism.

[From Discussion, p. 13]:

The A β binding region of PrP^C, which is comprised of two short amino acid motifs within the intrinsically disordered N-terminal domain¹⁰, is structurally unrelated to the A β binding regions of Fc γ RIIb and LILRB2, which are composed of one or two immunoglobulin domains, respectively^{11,12}. It remains to be determined how structurally diverse receptors and chaperones selectively recognize localized binding sites on A β fibrils and oligomers³⁵. One possibility is that A β binding induces conformational changes in the receptors that enhance their affinity for specific structural features on these assemblies. Consistent with this idea, NMR studies show that A β oligomer binding induces substantial conformational changes in several regions of PrP^C^{59,74}.

7) How would the authors reconcile their data about the inhibitory effect of PrPC on A β polymerization with the presence of senile plaques in the Alzheimer's brain as A β plaques originate from the previous polymerization of A β peptides and subsequent deposition?

As discussed above (point #4), the major effects of PrP^C *in vivo* are likely to occur on the cell surface, and to result in trapping of oligomers and protofibrils on the neuronal membrane. Thus, the bulk of the A β deposited in the extracellular space, including in plaques, would likely not come into contact with PrP^C, and in any case is probably present in a large excess over the amount of cell-associated PrP^C. Given its inhibitory effect on A β polymerization, PrP^C could contribute to increasing the amount of soluble, neurotoxic A β species (oligomers, protofibrils), which are considered to represent a pool distinct from plaque-associated A β (see Liu, Reed et al., *Cell Rep.* 11: 1760-1771; 2015).

Reviewer #2 (Remarks to the Author):

The authors report super-resolution analyses of interactions between A-beta assemblies, PrPC and two other receptors for A-beta assemblies. They demonstrate clearly the polarized interaction of PrPC with the more rapidly growing ends of A-beta oligomers, protofibrils and fibrils. This interaction blocks the further growth of these assemblies. The findings are novel and relevant to understanding the how pathological A-beta species interact with receptors that mediate neurotoxicity in the context of Alzheimer's disease. The presentation is clear and illuminating. The data are visually stunning, mechanistically revealing, and supportive of the testing therapeutic strategies that target these interactions (with the caveat noted below). I found little to criticize. However, regarding therapeutic strategies, I am not sure that it is easy to predict the *in vivo* consequences of blocking the binding of A-beta species with their receptors. On the face of it, as the authors argue, one might expect an immediate reduction in neurotoxic signal transduction induced by existing A-beta assemblies. On the other hand, if receptor interactions block the growth of pathological A-beta species as the authors have shown, then blocking the interactions might promote the accumulation of A-beta multimers. Granted, the longer the fibrils the less toxic they appear to be per particle, but fibrils still have some toxicity, and if there are proteostatic mechanisms that fragment them, a treatment that blocks the ability of receptors such as PrPC to reduce A-beta fibrilization might have a detrimental effect. Of course, we won't know whether the potentially beneficial or detrimental effects of such a treatment would win the day until appropriate trials are done.

We thank the reviewer for his positive comments on the manuscript. We understand the point the reviewer is making. Based on our comments above (reviewer #1, point #7), we would argue that blocking the interaction of A β with PrP^C and/or other receptors (for instance by small molecules or antibodies), might not have a major effect on the bulk of A β deposited in the brain. However, this strategy might strongly reduce A β toxicity by preventing trapping of oligomers and protofibrils

on the neuronal surface, thereby blocking their activation of receptor-mediated toxic signaling. We have now elaborated on this point in the last paragraph of the Discussion section.

[From Discussion, p. 14]:

Current therapies for AD are focused primarily on lowering levels of A β , either by inhibiting its synthesis or enhancing its degradation⁸⁴. These therapies have met with little success in recent clinical trials. The results presented here raise the possibility of a novel therapeutic approach based on blocking interactions of neurotoxic forms of A β with its cellular receptors, thereby reducing the trapping of toxic oligomers and protofibrils on the neuronal surface, and inhibiting activation of downstream signaling pathways engaged by these receptors. Indeed, there is evidence that PrP^C-directed ligands (small molecules and antibodies) can have positive therapeutic effects in AD mouse models^{31,32,85}. We suggest that it will be possible to fine-tune this approach, based our observation that there is a common structural interface on A β fibril ends, as well as on neurotoxic protofibrils and oligomers, that is recognized by multiple A β receptors. By defining this interface at the atomic level, it may be feasible to design small molecules that block interaction of neurotoxic A β species with all of these receptors simultaneously, thereby providing a highly efficient AD therapeutic. This structural information may also inform creation of diagnostic reagents specific for neurotoxic forms of A β .

Reviewer #3 (Remarks to the Author):

1. The main question that comes to mind centers on the physiological relevance of this biochemical process observed in vitro. Does this effect of PrPc and other receptors on A β fibril formation in isolated highly simplified in vitro systems actually affect disease processes in the human brain? The μ M concentrations of PrPc and A β used may not be physiological in most areas of the brain, and PrPc and the other receptors are mainly cell surface bound rather than diffusing freely.

We have addressed this question above (Reviewer #1, point #4).

2. What is causing the toxic effect of the PrPc interaction with amyloid? Is it the fact that PrPc (or LILrB, etc.) prevents the growth of the long end of amyloid fibrils creating greater numbers of smaller, more toxic species that cause greater cellular damage? This could happen if soluble species of PrPc are circulating and disrupting fibrilization. Or is it simply that amyloid fibril binding the PrPc, LILrB or Fc γ RIIb activates various toxic intracellular signaling cascades, and the effect on amyloid fibrillization is just a consequence of preferential binding, but not really relevant to disease processes? Or may both pathways be disease relevant?

We would argue that both mechanisms are operative, and indeed are interrelated. As we have stated in the Discussion section, we hypothesize that PrP^C recognizes common structural determinants present on the growing ends of fibrils and protofibril ends, as well as on oligomers. Thus, PrP^C can capture and trap these neurotoxic forms on the neuronal surface, and could also potentially

contribute to their formation as A β polymerizes in proximity to the membrane surface. As stated above (Reviewer #1, point #4), we are planning SRM experiments to investigate how membrane-anchored PrP^C influences A β polymerization and the formation of A β oligomers on the neuronal surface. The bulk of A β deposition (in plaques, for example) probably occurs in the extracellular space, distant from any cellular source of PrP^C (Reviewer #1, point #7), although it is known that some soluble PrP^C is shed from the cell membrane (Linsenmeier et al., *Mol. Neurodegen.* 13:18, 2018), and could influence A β polymerization more widely. However, the majority of PrP^C in the brain is membrane-anchored, so this mechanism may make only a minor contribution.

3. Related to #1 above, cell culture or *in vivo* experiments should be performed to illustrate the physiological consequences of this biochemical mechanism. Perhaps a mutant PrPc that still binds amyloid fibrils and affects fibril growth, but has lost intracellular signaling capabilities could differentiate effects. Or amyloid oligomers could be added to primary cultures in the presence and absence of recombinant PrPc followed by a toxicity assay, such as measurement of dendritic spine density. If blocking fibrilization increases toxicity, then soluble PrPc would increase toxicity. Alternatively, if binding to an Abeta receptor caused toxicity, then soluble PrPc should be protective. It would also be informative to Abeta-treat PrPC^{-/-} neurons, which should be resistant to Abeta toxicity.

The reviewer raises several important points here. First, is the question of whether the effect of PrP^C *in vivo* is to mediate intracellular signaling pathways, and/or to enhance production of neurotoxic forms of A β (short fibrils, protofibrils, oligomers). As we have discussed above, both of these mechanisms would contribute to toxicity, and both could be operative (see point #2, above). As a way to distinguish these two mechanisms, the reviewer suggests "...using a mutant PrP^C that still binds amyloid fibrils and affects fibril growth, but has lost intracellular signaling capabilities...". In fact, we are planning to systematically dissect the signaling capabilities of PrP^C using just such as structure-function approach. We will introduce various mutant PrP^C constructs into PrP^C-null neurons and assess their ability to mediate A β oligomer toxicity using our spine retraction assay. However, we feel that these technically demanding experiments, which we have included in a recent grant application, are well beyond the scope of the present paper, and are properly the subject of a subsequent publication.

Second, the reviewer suggests testing how addition of recombinant PrP to A β affects its neurotoxic activity. We have performed two new experiments to implement the reviewer's suggestion, and the results are now presented in three new figures (Fig. 6 and Supplementary Figures S7 and S8). Our results actually verify the predictions of the reviewer: the presence of PrP during the polymerization reaction favors accumulation of short, highly neurotoxic fibrils, but that saturation of these fibrils with additional PrP reduces their toxic effect. Addition of soluble PrP to pre-formed ADDLs and protofibrils also reduces the neurotoxicity of these preparations. All of these neurotoxicity experiments are now collected into a new sub-section of the results, as shown below.

[From Results, p. 8]:

Effect of PrP on neurotoxicity of A β assemblies

Given the specific and spatially localized binding of PrP to A β oligomers and protofibrils, we asked whether this interaction played a role in the neurotoxic activity of these forms in a relevant biological system. We predicted that binding of exogenous, recombinant PrP to neurotoxic A β assemblies would block their ability to bind to PrP^C on the neuronal surface, and thereby reduce their ability to induce neurotoxic effects. To test this prediction, we assayed the neurotoxicity of several kinds of A β aggregates by measuring their ability to induce retraction of dendritic spines on cultured hippocampal neurons⁵¹. We and others have demonstrated that A β oligomers cause rapid changes in the morphology and function of dendritic spines, and that these synaptotoxic effects depend on expression of PrP^C by target neurons^{25,52}.

We found, first, that synaptotoxicity was correlated with the size of A β aggregate. ADDLs and 3-day protofibrils induced more spine retraction than 7-day protofibrils, (Supplementary Fig. S7A-F). These data confirm previous evidence that smaller oligomers are more toxic than fibrils⁵, and they are consistent with the idea that smaller assemblies display a higher molar concentration of binding sites for cell-surface receptors like PrP^C that transduce neurotoxic signals. We next incubated pre-formed ADDL and protofibril preparations with soluble, recombinant PrP prior to addition to neuronal cultures. We observed that pre-treatment with soluble PrP partially suppressed spine retraction by these A β preparations, and this effect was dependent on the concentration of PrP added (Supplementary Fig. S7G-J).

We then performed a second kind of experiment to investigate how the toxicity of A β is affected by polymerization in the presence of PrP. Since inclusion of PrP in the polymerization reaction results in the accumulation of large numbers of short A β fibrils (Fig. 1), we asked whether these short fibrils were neurotoxic. In these experiment, we first polymerized A β monomers (5 μ M) in the presence of increasing concentration of PrP (0, 0.1, 0.5 and 1 μ M) for 24 h (Fig. 6A). The A β aggregates that formed were then added to cell media at a final concentration of 500 nM (A β monomer-equivalents) and incubated for 24 h. We found that A β samples polymerized with PrP were highly neurotoxic, and their toxicity increased as the concentration of PrP present in the reaction increased (Fig. 6B-D), correlating with the presence of shorter and more numerous fibrils (Fig. 1). In addition, we observed that the toxicity of these short fibrils was blocked by addition of excess recombinant PrP (5 μ M) to the polymerization reaction, prior to dilution into tissue culture medium for treatment of the neurons (Fig 6E-F). This rescuing effect was also observed when fibrils were polymerized using a higher concentration of A β (20 μ M) (Supplementary Fig. S8). This experiment demonstrates that the presence of PrP during the polymerization reaction favors accumulation of short, highly neurotoxic fibrils, but that saturation of these fibrils with additional PrP reduces their toxic effect.

Taken together, both of these experiments demonstrate that addition of exogenous, recombinant PrP to several different A β assemblies significantly reduces their neurotoxic effect on cultured hippocampal neurons. This result is consistent with the idea that the added PrP competes with cellular PrP^C for binding to these A β species, and therefore supports the role of endogenous PrP^C as a receptor that mediates A β neurotoxicity.

Finally, the reviewer notes that “it would also be informative to A β -treat PrP^{C-/-} neurons, which should be resistant to A β toxicity”. We have actually already published this experiment, with the predicted result⁵¹.

Reviewer #4 (Remarks to the Author):

1) What is the aggregation state of the PrP added to the assay? Is it the monomeric protein or a preformed PrP aggregate? The authors need to show data characterizing the aggregation / oligomerization of PrP by itself under the assay conditions. The underlying hypothesis is that membrane-bound PrP monomers interact with the nascent fibril. However, whether this is the case in vitro, is not shown.

We start each experiment with freshly prepared PrP monomers (see Material and Methods). Although it is possible that some aggregation of PrP could occur during the subsequent incubations with A β , we now have a piece of experimental evidence that makes this scenario unlikely (new Fig. S4). This experiment, which utilized a 50:50 molar mixture of PrP molecules labeled with either one of two different fluorophores (PrP-AF555 and PrPAF488), was designed to test the stoichiometry of PrP binding to A β fibrils. However, it also provides data on the aggregation state of PrP that is free in solution and not bound to fibrils. We find that only 25 \pm 0.03% of the PrP clusters in each field that are not associated with fibrils contain both fluorophores, compared to 58 \pm 0.03% of the PrP clusters localized to fibril ends. This result indicates that while most PrP molecules are monomeric before fibril binding, two or more molecules of PrP can associate with each fibril end. This fact is mentioned at the end of the paragraph in the Results section that describes this experiment.

[From Results, p. 7]:

Finally, we wished to investigate the stoichiometry of PrP binding to fibrils ends. In order to determine whether each fibril end bound one or more molecules of PrP, we performed a triple-label experiment (Supplementary Fig. S4) in which A β -Cy5 monomers were incubated for 24 h in the presence of an equimolar (50:50) mixture of PrP labeled with either Alexa Fluor 488 (PrP-AF488) or Alexa Fluor 555 (PrP-AF555). Samples were then imaged with three-color SIM. In each image, we measured the number of A β -associated PrP clusters containing either AF488 or AF555, as well as the number of PrP clusters containing both fluorophores. We determined that 58 \pm 0.03% of the fibril-associated PrP clusters were labeled with both AF488 and AF555,

indicating that the majority of clusters contained more than one molecule of bound PrP. Several such clusters are indicated by arrows in Fig. S4A. These data indicate that most fibril ends bind more than one PrP molecule. We also found that only $25 \pm 0.03\%$ of the PrP clusters not associated with fibrils contained both labels, indicating that most unbound PrP molecules were monomeric. This result argues against artifactual aggregation of PrP during the course of the experiment.

2) Does membrane bound PrP on the cell membrane cap the growth of Abeta fibrils the same way the soluble recombinant protein does? Expression of GPI-less PrP *in vivo* produces a very different prion disease phenotype from normal membrane-bound PrP, so it is likely that the molecular properties of both forms are quite different.

We agree with the reviewer that the localization of PrP^C on the cell membrane is a key factor in assessing how it interacts with A β *in vivo*. We have addressed this question in response to Reviewer #1, point #4.

3) What are the binding constants of PrP to Abeta oligomers and fibrils, respectively? This is an important piece of data to judge whether the interaction observed *in vitro* is plausible under physiological conditions.

We estimated the binding affinity of PrP for A β fibrils and oligomers in a previous publication²². We fit thioflavin polymerization curves in the presence of different amounts of PrP to published differential equations describing the kinetics of A β polymerization. In this scheme, the data were best modeled by assuming that PrP specifically reduces k_+ , the rate constant for fibril elongation, and that it does so by binding to fibril ends with an equilibrium dissociation constant, K_d , of 47.6 nM. Using DELFIA, we estimated the K_d for PrP binding to A β oligomers to be <100 nM. These binding affinities are sufficiently strong to make it likely that the PrP-A β interactions observed *in vitro* are relevant to physiological conditions.

The authors then look at the evolution of Abeta oligomers (ADDL) into protofibrils and fibrils and find that PrP shifts the population towards shorter, more toxic species, most likely by the fibril end-capping mechanism characterized in Fig 3. This is a very nice result, but it raises a number of technical questions, especially about the discrimination between oligomers, protofibrils and fibrils and whether the aggregates size distribution under conditions of the toxicity assay is the same as it was *in vitro*:

4) How is a protofibril discriminated from an ADDL, and how is a protofibril distinguished from a fibril? It seems that the length distribution of protofibrils extends into objects of several micrometers, which is a typical length of fibrillar Abeta. Superresolution images of panels labeled as protofibrils show straight fibril-shaped objects. Typically, protofibrils are defined as worm-like, short linear assemblies that are distinct in diameter and shape from straight amyloid fibrils. The

data shown here give no indication that the object imaged are indeed protofibrils. This claim needs to be backed up by AFM or EM imaging and a clear criterion for distinguishing both types of assemblies needs to be established.

As suggested by the reviewer, we have used EM to compare the structure and length distribution of ADDLs, protofibrils 1 and 2, (New Fig. S5). The following paragraph was inserted into the main text.

[From Results, p. 7]:

We used a standard and widely used method^{48,50} to produce ADDLs and protofibrils, based on resuspending a dried film of A β peptide (evaporated from DMSO) in tissue culture medium, and incubating it for different periods of time. After 16 hrs of incubation the sample contains mostly small, oligomeric assemblies (ADDLs), while incubation for 3 days and 7 days results in increased numbers of protofibrils. First, we used EM to characterize the structures of the ADDL and protofibril preparations used in our experiments. (Supplementary Fig. S5). Consistent with many previously published studies⁴⁸, ADDLs consisted primarily of a heterogeneous population of globular and ellipsoid structures with the majority having a diameter of 5-15 nm, just at the resolution limit of dSTORM. In protofibril preparations formed by incubation of ADDLs for 3 days, short, worm-like assemblies with a mean length of 16.0 nm began to appear. After 7 days of incubation, these assemblies became longer, with a mean size of 49.2 nm. In contrast, mature fibrils polymerized directly from monomeric A β for 24 hrs were much longer (see Fig. 1 and 2). The worm-like structures observed in the two protofibril preparations displayed an irregular surface, in contrast to the smooth surface of mature fibrils. The morphology of the ADDL preparations incubated for 3 and 7 days is consistent with published images of protofibrils²¹.

5) S4 the length distribution of protofibrils in the panel B of Fig S4 look very heterogeneous, which is also reflected in the distribution of panel E. However, panel D shows a narrow distribution with tight error bars. Please explain this apparent discrepancy.

These preparations are indeed heterogeneous, especially during the first 24 h of incubation. However, the bars in panel D represent mean \pm standard error for a large number of aggregates (sometimes more than 1,000 objects were detected). The standard error for such a large population is expected to be very small.

6) Similarly, panels F-I in S5 show a distribution of spherical objects (presumably ADDL) and fibril-shaped objects of varying length. It is not clear to me how protofibrils fit into this picture. The data could easily be explained by a conversion of oligomers (ADDL) into fibrils. It has been well established that oligomers are the main toxic Abeta species, so that their depletion would explain the observed reduction in toxicity that matches the disappearance of oligomeric species and the growth of fibrils.

Our ADDL and protofibril preparations have now been characterized morphologically by EM, as discussed under point #4, above (see new Fig. S5). These preparations are relatively stable, and evolve very slowly over the course of 7 days (in contrast to fresh A β monomers, which polymerize rapidly over 24 hours; see Figs. 2 and S2). As pointed out above, the morphology of the structures in the ADDL preparations incubated for 3 or 7 days indicates that they contain protofibrils, rather than fibrils. Thus, the relative neurotoxicity of ADDLs > protofibrils (3-day) > protofibrils (7-day) > mature fibrils correlates with size and intrinsic structure, which we postulate is related to their ability to bind cell-surface receptors like PrP^C (see Fig. S7).

7) The length distribution of fibrils and 3 d protofibrils used in toxicity assays is missing.

The length distributions of ADDLs, 3-day protofibrils, and 7-day protofibrils, measured by EM, are now shown in Fig. S5. Fig. 1 already showed the length distribution of fibrils measured by SIM.

8) The method section states that for toxicity assays, neurons were treated with 500 mM Abeta. This is clearly wrong as the Abeta peptide is not soluble at 0.5M concentration. Did the authors mean 500 μ M? Even that would be a huge concentration, which would probably lead to uncontrolled aggregation of the peptide in the cell culture medium. The authors need to show that the size distribution of Abeta aggregates encountered by the neurons is actually the same that they characterized in their in vitro aggregation assays.

We thank the reviewer for pointing out this typographical error, which we have now corrected. Neurons were treated with 500 nM (not 500 mM) monomer-equivalent of A β . Since the ADDL and protofibril preparations used for the neurotoxicity assays were prepared in the same way as for SRM and EM (by incubation of A β in tissue culture medium), we can be reasonably confident that neurons were exposed to the same structures as were imaged microscopically.

9) Minor point: Scale bars are missing in S4panels B-C, scale bar is not defined in length in panel A. Scale bars missing in all panels of Fig S5 neuronal images.

We have now included this information in both figures.

Finally, the authors analyze the effect of the two membrane receptors Fc γ RIIb and LILRB2 on Abeta fibril growth. Analysis of bulk aggregation finds that both proteins (or more specifically their extracellular domains) inhibit aggregation in vitro. Both protein also shift fibril size distribution towards shorter species, similar to PrP. This is a very suggestive result. However, it raises the same conceptual and technical issues just discussed for PrP:

10) The authors need to show that the proteins actually interact with the growing fibril ends of Abeta.

We agree with the reviewer that it would be desirable to include SRM experiments localizing FcγRIIb and LILRB2 on Aβ fibrils. Unfortunately, the super-resolution microscope we use is located in a Harvard core facility (Center for Biological Imaging), which was closed during the COVID lock-down period, and has been operating with only a limited capacity since then. The same is true for other core facilities in the city.

11) What is the aggregation state of the proteins?

We acquired these two proteins from commercial sources (Novoprotein and R&D Systems), and the quality control information from the companies indicates that they are monomeric. There is no evidence that these proteins normally multimerize.

12) How can we be sure that the membrane-associated native form of the proteins has the same effect on Abeta aggregation as the extracellular domain fragment?

We would address this concern using arguments analogous to those we offered in response to Reviewer #1, point #4 with regard to PrP^C. There are literature studies that support a specific physical interaction between Aβ aggregates and both FcγRIIb and LILRB2 on the surface of cultured cells, as well as *in vivo* in brain tissue from AD patients and transgenic mice. We are planning our own SRM experiments to investigate how these membrane-anchored receptors influence Aβ polymerization and the formation of Aβ oligomers on the neuronal surface, but these will be the subject of a subsequent paper.

Reviewers' comments:

Reviewer #1 (Remarks to the Author):

I commend the authors for their spirit of responsiveness to constructive criticism and their polite and thorough responses. The authors have completed a substantial amount of additional work to address my concerns regarding the stoichiometry of interaction between recombinant PrP and the fast-growing end of A β fibrils, or the binding of PrP restricted to the fast-growing end of A β fibrils.

Nevertheless, although the authors integrated additional references that document the interaction of cellular prion protein (PrPC) present at the plasma membrane of neurons with different A β assemblies, the paper by Amin et al. still suffers from a lack of experimental data showing a real, physiological, inhibitory effect of PrPC expressed by neurons or other brain cells on A β polymerization and connection with A β neurotoxicity. Such a PrPC role thus remains highly speculative. No experiment based on primary hippocampal neurons expressing or not PrPC has been proposed to address the potential in vivo inhibitory effect of PrPC on A β polymerization.

Moreover, ex vivo data using primary cultures of hippocampal neurons and preparations of A β (ADDLs, protofibrils, fibrils) pre-treated or not with recombinant PrP do not permit to convincingly establish the link between the PrP-mediated inhibition of A β polymerization and the toxicity of smaller A β assemblies. In fig. S7, the authors show that ADDLs and A β protofibrils are neurotoxic by themselves. The interaction of these A β assemblies with low concentrations (0.1 to 1 μ M) of recombinant PrP (which inhibits the growth of protofibrils - Fig. S6 - and thereby would increase the neurotoxicity of A β species according to the author's postulation) exerts a protective effect, that is, reduces the synaptotoxicity of A β species pre-treated with recombinant PrP. By contrast, the pre-incubation of A β fibrils with the same concentrations of recombinant PrP enhances A β synaptotoxicity (Fig. 6), while reducing A β size and increasing the number of smaller A β entities (Figs. 1 and 2). By manipulating the interaction of A β species with recombinant PrP, it is likely the authors change the number of free A β ends that would be recognized by cell surface PrPC, but also by other A β receptors, which might be more critical for the initiation of neurotoxic signals than the inhibition of A β polymerization if any.

Reviewer #2 (Remarks to the Author):

The authors have adequately addressed my query.

Reviewer #3 (Remarks to the Author):

The authors have adequately revised the manuscript.

Reviewer #4 (Remarks to the Author):

In their revised manuscript the authors have added new TEM and fluorescence data, which convincingly address my main two points of criticism with regard to the identity of different aggregate species and with regard to the aggregation state of PrP. The dual-label mixing

approach in an elegant approach to solving the difficult problem of quantifying oligomer states in fluorescence data.

The manuscript presents an elegant piece of work that addresses a central point of the amyloid self-assembly and toxicity mechanism, which will be well appreciated by the field.

MS ID#: NCOMMS-19-34520B

MS TITLE: A β receptors specifically recognize molecular features displayed by fibril ends and neurotoxic oligomers

AUTHORS: Ladan Amin and David A. Harris

Reviewer #1 (Remarks to the Author):

I commend the authors for their spirit of responsiveness to constructive criticism and their polite and thorough responses. The authors have completed a substantial amount of additional work to address my concerns regarding the stoichiometry of interaction between recombinant PrP and the fast-growing end of A β fibrils, or the binding of PrP restricted to the fast-growing end of A β fibrils.

Nevertheless, although the authors integrated additional references that document the interaction of cellular prion protein (PrPC) present at the plasma membrane of neurons with different A β assemblies, the paper by Amin et al. still suffers from a lack of experimental data showing a real, physiological, inhibitory effect of PrPC expressed by neurons or other brain cells on A β polymerization and connection with A β neurotoxicity. Such a PrPC role thus remains highly speculative. No experiment based on primary hippocampal neurons expressing or not PrPC has been proposed to address the potential in vivo inhibitory effect of PrPC on A β polymerization.

Demonstrating that PrP^C inhibits A β polymerization on the surface of intact neurons or other cells is extremely challenging from a technical standpoint. We did, in fact, attempt to address this concern experimentally after we received the initial reviews of the manuscript. In these pilot experiments, we added monomeric A β to the culture medium bathing HEK cells, which either over-expressed or lacked PrP^C, and monitored polymerization by ThT fluorescence. However, the results were highly variable, and the viability of the cells became compromised during the prolonged incubation periods.

It is clear that the best way to assess the effects of cell-associated PrP^C is using super-resolution or single-particle microscopy of live cells exposed to fluorescently labeled A β , as was done by Ganzinger et al. (2014) (cited as ref. 23 in the new paragraph in the Discussion section). We are planning to undertake such experiments, but feel that these are properly the subject of a subsequent paper, given their technical difficulty. Moreover, we continue to be hampered by limited access to the super-resolution imaging facility at Harvard due to COVID restrictions. We have no doubt that once these experiments can be undertaken, they will yield a wealth of new information that could easily constitute a separate paper, including the orientation and distribution of A β aggregates assembled on or trapped near the surface of neurons (e.g., Are they attached by their ends?; Do

they co-localize with dendritic spines?), differences between different types of neuron and between neurons and glial cells, and the effect of anti-A β therapeutic agents.

In light of all these considerations, we felt that the best strategy to address the *in vivo* relevance of our results within the context of our manuscript was to devote a separate paragraph of the Discussion to the subject, citing ref. 23, as well as biochemical evidence that PrP^C forms complexes with neurotoxic A β species in brain tissue (refs. 74, 80-82). We note that reviewers #3 and #4 had raised a similar concern, but were apparently satisfied with the new paragraph we included, stating that the revised manuscript now adequately addressed all of their criticisms.

To acknowledge the limitation imposed by not providing our own data on A β /PrP^C interaction in a cellular context, we have inserted the following sentence into the Discussion:

[From Discussion, p. 14]:

Experiments using super-resolution or single-particle microscopy of live cells will be required to address definitively whether glycosylated, GPI-anchored PrP^C on the cell surface interacts differently with A β than recombinant PrP.

Reviewer #1 (Remarks to the Author, continued):

Moreover, ex vivo data using primary cultures of hippocampal neurons and preparations of A β (ADDLs, protofibrils, fibrils) pre-treated or not with recombinant PrP do not permit to convincingly establish the link between the PrP-mediated inhibition of A β polymerization and the toxicity of smaller A β assemblies. In fig. S7, the authors show that ADDLs and A β protofibrils are neurotoxic by themselves. The interaction of these A β assemblies with low concentrations (0.1 to 1 μ M) of recombinant PrP (which inhibits the growth of protofibrils - Fig. S6 - and thereby would increase the neurotoxicity of A β species according to the author's postulation) exerts a protective effect, that is, reduces the synaptotoxicity of A β species pre-treated with recombinant PrP. By contrast, the pre-incubation of A β fibrils with the same concentrations of recombinant PrP enhances A β synaptotoxicity (Fig. 6), while reducing A β size and increasing the number of smaller A β entities (Figs. 1 and 2). By manipulating the interaction of A β species with recombinant PrP, it is likely the authors change the number of free A β ends that would be recognized by cell surface PrP^C, but also by other A β receptors, which might be more critical for the initiation of neurotoxic signals than the inhibition of A β polymerization if any.

This comment reflects an apparent misunderstanding of how the experiments in Figs. 6 and S7 were performed. The A β aggregates used in Fig. S7 were pre-formed ADDLs and protofibrils that had been assembled in the absence of PrP, and that were then incubated with different

concentrations of PrP for 10 minutes prior to a 40-fold dilution into neuronal culture medium. This experiment demonstrated that pre-treatment of these A β preparations with soluble PrP partially suppresses dendritic spine retraction, and this effect is dependent on the concentration of PrP (Fig. S7, panel J). This experiment was not designed to test whether PrP alters the assembly process of the A β aggregates.

In contrast, in Fig. 6, A β was polymerized in the presence of PrP for 24 hours. The resulting fibrils, which are shorter and more numerous than control fibrils (see Figs. 1 and 2), were then either diluted directly into culture medium, or were incubated with extra recombinant PrP for 10 minutes prior to dilution into culture medium. This experiments demonstrated that the shorter A β fibrils formed during polymerization with PrP were highly neurotoxic, and their toxicity increased as the concentration of PrP increased (Fig. 6D), correlating with the formation of progressively shorter and more numerous fibrils (Figs. 1 and 2). Moreover, the toxicity of these short fibrils was blocked by incubating them with extra recombinant PrP prior to dilution into tissue culture medium (Fig. 6F). The latter observation is completely consistent with what we show in Fig. S7, and indicates that recombinant PrP added just before treatment of neuronal cultures competes with cellular PrP^C for binding to neurotoxic A β species, and therefore supports the role of endogenous PrP^C as a receptor that mediates A β neurotoxicity.

Reviewers' comments:

Reviewer #1 (Remarks to the Author):

Although I still believe that the phenomenon observed in vitro with recombinant PrP has to be established in a cellular context, the manuscript was revised adequately and clarifications were done.